# Graphcode: Learning from multiparameter persistent homology using graph neural networks

**Michael Kerber**[*]
Institute of Geometry
Graz University of Technology
kerber@tugraz.at

**Florian Russold**[*]
Institute of Geometry
Graz University of Technology
russold@tugraz.at

## Abstract

We introduce graphcodes, a novel multi-scale summary of the topological properties of a dataset that is based on the well-established theory of persistent homology. Graphcodes handle datasets that are filtered along two real-valued scale parameters. Such multi-parameter topological summaries are usually based on complicated theoretical foundations and difficult to compute; in contrast, graphcodes yield an informative and interpretable summary and can be computed as efficient as one-parameter summaries. Moreover, a graphcode is simply an embedded graph and can therefore be readily integrated in machine learning pipelines using graph neural networks. We describe such a pipeline and demonstrate that graphcodes achieve better classification accuracy than state-of-the-art approaches on various datasets.

## 1 Introduction

A quote attributed to Gunnar Carlsson says "Data has shape and shape has meaning". Topological data analysis (TDA) is concerned with studying the shape, or more precisely the topological and geometric properties of data. One of the most prominent tools to quantify and extract topological and geometric information from a dataset is persistent homology. The idea is to represent a dataset on multiple scales through a nested sequence of spaces, usually simplicial complexes for computations, and to measure how topological features like connected components, holes or voids appear and disappear when traversing that nested sequence. This information can succinctly be represented through a *barcode*, or equivalently a *persistence diagram*, which capture for every topological feature its lifetime along the scale axis. Persistent homology has been successfully applied in a wealth of application areas [14, 23, 28, 30, 31, 36], often in combination with Machine Learning methods – see the recent survey [22] for a comprehensive overview.

A shortcoming of classical persistent homology is that it is bound to a single parameter, whereas data often is represented along several independent scale axes (e.g., think of RGB images which have three color channels along which the image can be considered). To get a barcode, one is forced to chose fixed scales for all but one scale parameters. The extension to *multi-parameter persistent homology* [7, 9] avoids to make such choices. Similar to the one-parameter setup, the data is represented in a nested multi-dimensional grid of spaces and the evolution of topological features in this grid is analyzed. Unfortunately, a succinct representation as a barcode is not possible in this extension, which makes the theory and algorithmic treatment more involved. Nevertheless, the quest of how to use informative summaries in multi-parameter persistence is an active field of contemporary research. A common theme in this context is *vectorization*, meaning that some (partial) topological information is extracted from the dataset and transformed into a high-dimensional vector suitable for machine learning pipelines.

---

[*]Equal contribution.

38th Conference on Neural Information Processing Systems (NeurIPS 2024).

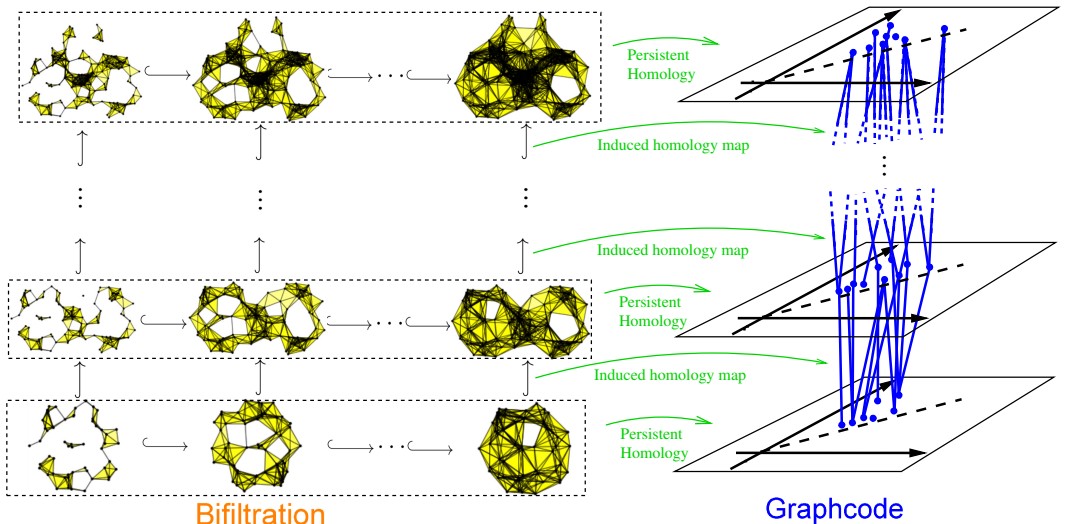

Figure 1: Schematic overview of our approach.

**Contribution.** We introduce *graphcodes*, a novel representation of the homology of a dataset that is filtered along two scale parameters. The idea, depicted in Figure 1, is to consider one-parameter slices of the dataset by fixing one parameter, obtaining a stack of persistence diagrams. We define a map between two consecutive diagrams in this stack, resulting in a bipartite graph connecting these diagrams. The graphcode is the union of these bipartite graphs over all consecutive pairs.

Since the maps connecting diagrams depend on a choice of basis for each persistence diagram, the graphcode is not a topological invariant. Nevertheless, the graphcode is a combinatorial description whose features are easy to interpret and which permits, for fixed bases per diagram, a complete reconstruction of the persistence module induced by the bifiltration. Moreover, the structure as an (embedded) graph in $\mathbb{R}^3$ admits a direct integration of graphcodes into machine learning pipelines via graph neural network architectures. In that way, graphcodes avoid the vectorization step of other homology-enhanced learning pipelines which often require more parameter choices and are sometimes slow in practice. In contrast, we describe an efficient algorithm to compute the graphcode of a bifiltered simplicial complex which essentially computes all required information through a single out-of-order matrix reduction of the boundary matrix of the entire complex. While the worst-case complexity is cubic, the practical performance is closer to linear for realistic datasets [5, 29].

We demonstrate how graphcodes facilitate classification tasks. For that, we implemented a machine learning pipeline that feeds the graphcodes into a simple graph neural network pipeline. On graph datasets used in related works on multi-parameter persistent learning, our approach shows a comparable classification quality. As a proof of concept, we also created a synthesized dataset of point clouds in $\mathbb{R}^2$ that contain a number of densely sampled disks and annuli plus some uniform noise. Clearly, topological classifiers are well-suited for such data. In our experiments, graphcodes outperform related methods on this type of data in terms of accuracy. At the same time, graphcodes are faster computed than all alternative topological descriptors, sometimes by several orders of magnitude. We also demonstrate that graphcodes perform better than other topological methods on two further datasets, established in related work on TDA, which consist of samples from different random point processes and orbits generated by a dynamical system, respectively.

**Related work.** Our method can be viewed as a generalization of PersLay [11] to the two-parameter case. PersLay is a neural network layer that enables vectorization-free learning from one-parameter persistent homology. It uses a deep set [37] architecture to directly take persistence diagrams as input. A conceptually simpler generalization of PersLay would consist of only using the union of persistence diagrams of the one-parameter slices, that is, the graphcode without connecting edges. We show in our experiments, however, that the edges improve the accuracy in the two-parameter case.

Most of the previous methods used in applications are based on transforming persistence diagrams into vectors in Euclidian space, or other data structures suitable for machine learning. Examples are

persistence landscapes [8], persistence images [1], or scale-space kernels [32]. These vectorization methods for one-parameter persistence modules have been generalized in various forms to the two-parameter case [10, 16, 18, 26, 35]. The difference in the two-parameter case is that the vectorizations are not based on a complete invariant like the persistence diagram but on weaker invariants like the rank-invariant, generalized rank-invariant or the signed barcode. Hence, these vectorizations capture the persistent homology only partially. Moreover, even this partial information is often times computationally expensive. In contrast, our method avoids to compute a direct vectorization, although we point out that a vectorization is implicitly computed eventually within the graph neural network architecture. To our knowledge there exists no other method that allows to feed a complete representation of two-parameter persistent homology into a machine learning pipeline.

Our approach also resembles persistent vineyards [17] in the sense that a two-parameter process is considered as a dynamic 1-parameter process and the evolution of persistence diagrams is analyzed. Indeed, vineyards produce a layered graph of persistence diagrams just as graphcodes (see Fig VIII.6 in [19]), but they operate in a different setting where the simplicial complex is fixed throughout the process and only the order of simplices changes, whereas graphcodes rely on bifiltered simplicial complex data that only increases along both axis. Most standard constructions of multi-parameter persistence yield such a bifiltered complex and graphcodes are more applicable in this situation.

Generating bifiltered simplicial complexes out of point cloud data is computationally expensive and an active area of research. In the context of the aforementioned two-dimensional point clouds that we analyze with graphcodes, we heavily rely on *sublevel Delaunay bifiltrations* which were introduced very recently by Alonso et al. [2]. That algorithm (and its implementation) render the two-parameter analysis of such point clouds possible in machine learning contexts, partially explaining why previous methods have only tested their approaches on very small point cloud data, if at all.

**Outline.** We review basic notions of persistent homology in Section 2 and define graphcodes in Section 3 based on these definitions. We decided for a "down-to-earth" approach, defining graphcodes in terms of cycle bases in simplicial complexes to keep concepts concrete and relatable to geometric constructions for the benefit of readers that are not too familiar with the algebraic foundations of persistent homology. Moreover, this treatment simplifies the algorithmic description to compute graphcodes in Section 4. We explain the machine learning architecture based on graphcodes in Section 5 and report on our experimental results in Section 6. We conclude in Section 7.

## 2   Persistent homology

We will use the following standard notions from simplicial homology. For readers not familar with these concepts, we provide a summary in Appendix A. For a more in-depth treatment see, for instance, the textbook by Edelsbrunner and Harer [19].

For an (abstract) simplicial complex $K$ and $p \geq 0$ an integer, let $C_p(K)$ denote its $p$-th chain group with $\mathbb{Z}_2$ coefficients (which is, in fact, a vector space) and $\partial_p : C_p(K) \to C_{p-1}(K)$ the boundary map (see also Figure 2 (left)). Let $Z_p(K)$ be the kernel of $\partial_p$. Elements of $Z_p(K)$ are called $p$-cycles. $B_p(K)$ is the image of $\partial_{p+1}$, and its elements are called $p$-boundaries. The quotient $Z_p(K)/B_p(K)$ is called the $p$-th homology group $H_p(K)$, whose elements are denoted by $[\alpha]_K$ with $\alpha$ a $p$-cycle, or just $[\alpha]$ if the underlying complex is clear from context. See Figure 2 (right) for an illustration of these concepts.

We call a basis $(z_1, \ldots, z_m)$ of $Z_p(K)$ *(boundary-)consistent* if there exists some $\ell \geq 0$ such that $(z_1, \ldots, z_\ell)$ is a basis of $B_p(K)$. In this case, $[z_{\ell+1}], \ldots, [z_m]$ is a basis of $H_p(K)$. A consistent basis can be obtained by first determining a basis of $B_p(K)$ and completing it to a basis of $Z_p(K)$. Clearly, $Z_p(K)$ usually has many consistent bases.

**Homology maps.** Let $L \subseteq K$ be a subcomplex. The inclusion map from $L$ to $K$ maps $p$-cycles of $L$ to $p$-cycles of $K$ and $p$-boundaries of $L$ to $p$-boundaries of $K$. It follows that for every $p$, the map $i_* : H_p(L) \to H_p(K), [\alpha]_L \mapsto [\alpha]_K$ is a well-defined map between homology groups. This map is a major object of study in topological data analysis, as it contains the information how the topological features (i.e., the homology classes) evolve when we embed them on a larger scale (i.e., a larger complex).

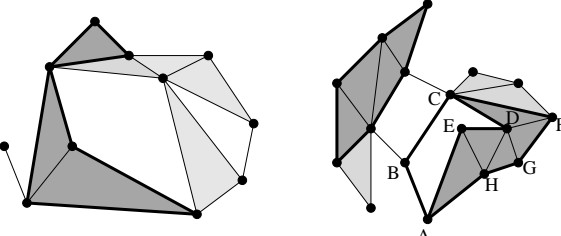

Figure 2: Left: A simplicial complex with 11 0-simplices, 19 1-simplices and 7 2-simplices. A 2-chain consisting of three 2-simplices is marked with darker color, and its boundary, a collection of 7 1-simplices is displayed in thick.

Right: The 1-cycle marked in thick on the left is also a 1-boundary, since it is the image of the boundary operator under the 4 marked 2-simplices. On the right, the 1-cycle $\alpha$ going along the path $ABCDE$ is not a 1-boundary; therefore it is a generator of an homology class $[\alpha]$ of $H_1(K)$. Likewise, the 1-cycle $\alpha'$ going along $ABCFGH$ is not a 1-boundary neither. Furthermore, $[\alpha'] = [\alpha]$ since the sum $\alpha + \alpha'$ is the 1-cycle given by the path $AEDCFGH$, which is a 1-boundary because of the 5 marked 2-simplices. Hence, $\alpha$ and $\alpha'$ represent the same homology class which is characterized by looping aroung the same hole in $K$.

Being a linear map, $i_*$ can be represented as a matrix with $\mathbb{Z}_2$-coefficients. It is instructive to think about how to create this matrix: Assuming having fixed consistent bases $A_L$ for $Z_p(L)$ and $A_K$ for $Z_p(K)$ and considering a basis element $[\alpha]_L$ of $H_p(L)$, we write the $p$-cycle $\alpha$ as a linear combination with respect to the basis $A_K$. We can ignore summands that correspond to basis elements of $B_p(K)$, and the remaining entries yield the image of $[\alpha]_L$ in $H_p(K)$, and thus one column of the matrix of $i_*$.

Alternatively, $i_*$ takes a diagonal form when picking suitable consistent bases $A_L$ and $A_K$. We can do that as follows: We start with a basis $A'$ of $Z_p(L) \cap B_p(K)$, the set of $p$-cycles in $L$ that become "trivial" when included in $K$. This subspace contains $B_p(L)$ and we can choose $A'$ such that it starts with a basis of $B_p(L)$, followed by other vectors. Now we extend $A'$ to a basis $A_L$ of $Z_p(L)$ which is consistent by the choice of $A'$. Since $Z_p(L)$ is a subspace of $Z_p(K)$, we can extend $A_L$ to a basis $A_K$ of $Z_p(K)$. Importantly, we can do so such that $A_K$ is consistent, because the sub-basis $A'$ maps to $B_p(K)$ and $A_L \setminus A'$ does not, so we can first extend $A'$ to a basis of $B_p(K)$, ensuring consistency, and then extend to a full bais of $Z_p(K)$. In this way, considering a homology class $[\alpha]$ with $\alpha$ a basis element of $A_L$, $\alpha$ is also a basis element of $A_K$, and the map $i_*$ indeed takes diagonal form.

**Filtrations and barcodes.** A *filtration* of a simplicial complex $K$ is a nested sequence
$$K_1 \hookrightarrow K_2 \hookrightarrow \cdots \hookrightarrow K_n = K$$
of subcomplexes of $K$. Applying homology, we obtain vector spaces $H_p(K_i)$ and linear maps $H_p(K_i) \to H_p(K_j)$ whenever $i \leq j$. This data is called a *persistence module*. For simplicity, we assume that $H_p(K)$ is the trivial vector space, implying that all $p$-cycles eventually become $p$-boundaries.

It is possible to iterate the above construction for a single inclusion to obtain consistent bases $A_i$ for $Z_p(K_i)$ such that all maps $H_p(K_i) \to H_p(K_j)$ have diagonal form. Equivalently, there is one basis of $Z_p(K)$ that contains consistent bases for all subcomplexes $K_1, \ldots, K_n$. To make this precise, we first observe that for every $\alpha \in Z_p(K)$, there is a minimal $i$ such that $\alpha \in Z_p(K_\ell)$ for all $\ell \geq i$. This is called the *birth index* of $\alpha$. Moreover, there is a minimal $j$ such that $\alpha \in B_p(K_\ell)$ for all $\ell \geq j$. This is called the *death index* of $\alpha$. By our simplifying assumptions that $H_p(K)$ is trivial, every $\alpha$ indeed has a well-defined finite death index. The half-open interval $[i, j)$ consisting of birth and death index is called the *bar* of $\alpha$. We say that $\alpha$ is *already born* at index $\ell$ if its birth index is at most $\ell$. We call $\alpha$ *alive* at $\ell$ if alpha is already born at $\ell$ and $\ell$ is strictly smaller than the death index of $\alpha$; otherwise we call it *dead* at $\ell$.

**Definition 2.1.** For a filtration of $K$ as above, a *barcode basis* is a basis $A$ of $Z_p(K)$ where each basis element is attached its bar such that the following property holds: For every $i$, the elements of $A$ dead at $i$ form a basis of $B_p(K_i)$, and the elements of $A$ already born at $i$ form a basis of $Z_p(K_i)$. In particular, these cycles form a consistent basis of $Z_p(K_i)$.

See Figure 3 (left) for an illustration. The collection of bars of a barcode basis is called the *barcode of $K$*. Indeed, while there is no unique barcode basis for $K$, they all give rise to the same barcode.

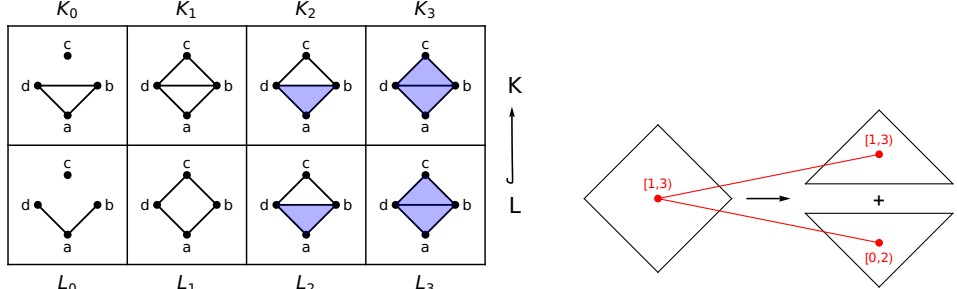

Figure 3: Left, lower row: $Z_1(L)$ is generated by the cycles $abcd$ and $abd$. They form a barcode basis, with attached bars $[1,3)$ and $[2,2)$, respectively. Note that also $abd$ and $bcd$ form a basis of $Z_1(L)$, but that is not a barcode basis as none of these cycles is already born at $L_1$, so they do not induce a basis of $Z_1(L_1)$. Left, upper row: Here, $abd$ and $bcd$ form a barcode basis with attached bars $[0,2)$ and $[1,3)$, respectively, and $abd$ and $abcd$ as well (with identical barcode).

Right: Choosing the basis $abcd$, $abd$ for $Z_1(L)$ and $abd$ and $bcd$ for $Z_1(K)$, we have $abcd = abd + bcd$, hence the cycle $abcd$ has two outgoing edges, to both basis elements in $K$. We ignore the basis vector $abd$ of $L$ in the figure, since its birth and death index coincide, so the corresponding feature has persistence zero.

This barcode (or the equivalent *persistence diagram*, where a bar $[i,j)$ is interpreted as a point $(i,j)$ in the plane) yields a topological summary of the filtration, revealing what topological features are active on which ranges of scale. Therefore, barcodes are a suitable (discrete) proxy for a dataset and are heavily used in applications.

The concept of a barcode basis enhances the barcode with a consistent choice of representative cycle for every bar. In practice, this extra information is obtained with no additional computation costs because the standard algorithm to compute barcodes computes a barcode basis as a by-product.

## 3 Graphcodes

We now consider the case of two filtered complexes $L$, $K$ such that $L_i \subseteq K_i$ for all $i$:

$$
\begin{array}{ccccccc}
K_1 & \hookrightarrow & K_2 & \hookrightarrow & \cdots & \hookrightarrow & K_n = K \\
\Big\uparrow & & \Big\uparrow & & & & \Big\uparrow \\
L_1 & \hookrightarrow & L_2 & \hookrightarrow & \cdots & \hookrightarrow & L_n = L
\end{array}
\tag{1}
$$

Assume we have fixed barcode bases $A_L$ for $Z_p(L)$ and $A_K$ for $Z_p(K)$. The inclusion $L \subseteq K$ induces a linear map $\phi : Z_p(L) \to Z_p(K)$, mapping each element of $A_L$ to a linear combination of $A_K$ with $\mathbb{Z}_2$-coefficients, or equivalently, to a subset of $A_K$. The map $\phi$ can be represented as a bipartite graph over $A_L \sqcup A_K$. By furthermore replacing elements of $A_L$ and $A_K$ with their attached bars, we can interpret this graph as a graph between barcodes, which we call the *graphcode* of (1). See Figure 3 (right) for an illustration. We emphasize that the graphcode depends on the chosen barcode bases for $L$ and $K$, thus the graphcode is not unique, and not a topological invariant.

The bases $A_L$ and $A_K$ together with their graphcode are sufficient to recover the homology maps $\phi_i : H_p(L_i) \to H_p(K_i)$, induced by the inclusion $L_i \subseteq K_i$: Since a basis of $Z_p(L_i)$ is contained in $A_L$, $\phi$ restricts to a map $Z_p(L_i) \to Z_p(K_i)$, and it is not hard to see that this map equals the map induced by the inclusion $Z_p(L_i) \to Z_p(K_i)$. Moreover, the basis of $Z_p(L_i)$ within $A_L$ is simply determined by those elements in $A_L$ that are already born at $i$, by definition of the barcode basis. Within this basis, the homology class represented by cycles alive at $i$ form a basis of $H_p(L_i)$. The image of these classes under $\phi$ yields a linear combination of cycles in $Z_p(K_i)$ which are all already born, and removing the summands corresponding to dead cycles yields the image of $\phi$ in $H_p(K_i)$.

**Bifiltrations.** Assume our data is now a *bifiltered simplicial complex* written as

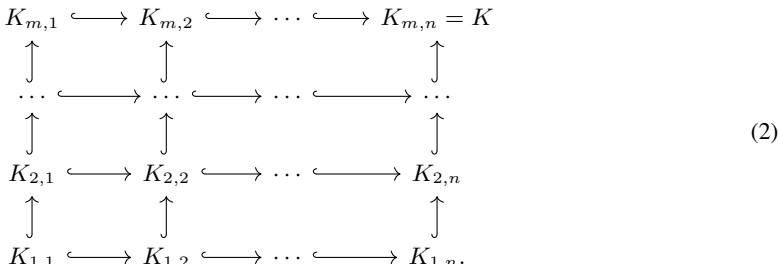

$$\tag{2}$$

Such a structure often times appears in applications where a dataset is analyzed through two different scales. An example is hierarchical clustering where the points are additionally filtered by an independent importance value.

We can iterate the idea from the last paragraph to a bifiltration in a straight-forward manner: Let $A_i$ be a barcode basis for the horizontal filtration $K_{i,1} \hookrightarrow K_{i,2} \hookrightarrow \cdots \hookrightarrow K_{i,n} = K_i$. With that bases fixed, there is a graphcode between the $i$-th and $(i+1)$-th horizonal filtration, and we define the union of these graphs as the *graphcode* of the bifiltration. The vertices of the graphcode are bars of the form $[b, d)$ that are attached to a basis $A_i$, and we can naturally draw the graphcode in $\mathbb{R}^3$ by mapping the vertex to $(b, d, i)$. This yields a layered graph in $\mathbb{R}^3$ with respect to the 3rd coordinate with edges only occurring between two consecutive layers. As discussed in Appendix D, graphcodes can be defined for arbitary two-parameter persistence modules. They can also be defined for arbitrary fields, in which case we obtain a graph that has not only node but also edge attributes.

## 4  Computation

The vertices and edges of a graphcode in homology dimension $p$ can be computed efficiently in $O(n^3)$ time where $n$ is the total number of simplices of dimension $p$ or $p + 1$. We expose the full algorithm in Appendix B in a self-contained way and only sketch the main ideas here for brevity.

First of all, it can be readily observed that the standard algorithm to compute persistence diagrams via matrix reduction yields a barcode basis in $O(n^3)$ time (see [19]). Doing so for every horizontal slice in (2) yields the vertices of the graphcode, and computing the edges between two consecutive slices can be reduced to solving a linear system via matrix reduction as well, resulting in $O(n^3)$ time as well for any two consecutive slices. This is not optimal though as it results in a total running time of $O(sn^3)$ with $s$ the number of horizontal slices.

To reduce further to cubic time, we perform an out-of-order matrix reduction, where the $(p+1)$-simplices are sorted with respect to their horizontal filtration value, but are added to the boundary matrix in the order of their vertical value. This reduction process, which still results in cubic runtime, yields a sequence of $n$ snapshots of reduced matrices that correspond to the barcode basis on every horizontal slice, and thus yields all vertices of the graphcode. The final observation is that with additional book-keeping when going from one snapshot to the next, we can track how the basis elements transform from one horizontal slice to the next and these changes encode which edges are present in the graphcode.

Finally, the practical performance can be further improved by reducing the size of the graphcode, by keeping $s$ small, by ignoring bars whose persistence is below a certain threshold, and by precomputing a minimal presentation instead of working with the simplicial input. See Appendix B for details.

## 5  Learning from graphcodes using graph neural networks

We describe our pipeline that exemplifies how graphcodes can be used in combination with graph neural networks (GNN's). The inputs are layered graphs with vertex attributes $[(b, d), i]$, with $[b, d)$ a bar of the barcode at the $i$-th layer. We can add further meaningful attributes like the additive $d - b$ and/or multiplicative $\frac{d}{b}$ persistence to the nodes to suggest the GNN that these might be important. Any graph neural network architecture can be used to learn from these topological summaries. We propose the architecture depicted in Figure 4. It starts with a sequence of graph attention (GAT) layers

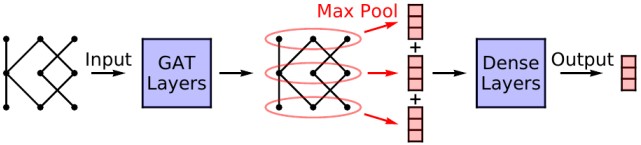

Figure 4: Neural network architecture for graphcodes.

[34] taking the graphcodes as input. The idea is that the network should learn to pay more attention to adjacent features with high persistence which are commonly interpreted as the topological signal. These layers are followed by a local max-pooling layer that performs max-pooling over all vertices in a common slice. Then we concatenate the vectors obtained from the local max-pooling over all slices and feed the resulting vector into a standard feed-forward neural network (Dense Layers).

If we remove all the edges from the graphcodes, this model can be viewed as a combination of multiple Perslay architectures [11], one for each slice of the bifiltration. In such a case, the model would implicitly learn for each barcode individually which bars are important for classification. Adding the edges, in turn, enhances this model as propagation between neighboring layers is possible: a bar that is connected to important bars in adjacent layers is more likely to be significant itself.

We also point out that the separate pooling by slices is crucial in our approach. It takes advantage of the additional information provided by the position of a slice in the graphcode. If we simply embed the entire graphcode in the plane by superimposing all persistence diagrams and do one global pooling, the outcome gets significantly worse.

## 6 Experiments

We have implemented the computation of graphcodes in a dedicated C++ library and the machine learning pipeline in Python. All the code for our experiments is available in the supplementary materials. The experiments were performed on an Ubuntu 23.04 workstation with NVIDIA GeForce RTX 3060 GPU and Intel Core i5-6600K CPU.

**Graph datasets.** We perform a series of experiments on graph classification, using a sample of TUDatasets, a collection of graph instances [27]. Following the approach in [10], we produce a bifiltration of graphs using the Heat Kernel Signature-Ricci Curvature bifiltration. From these bifiltrations, we compute the graphcodes (GC) and train a graph neural network as described in Section 5 to classify them. More details on these experiments can be found in Appendix C.1 and the supplementary materials. We compare the accuracy with multi-parameter persistence images (MP-I) [10], multi-parameter persistence kernels (MP-K) [18], multi-parameter persistence land-sacapes (MP-L) [35], generalized rank invariant landscapes (GRIL) [16] and multi-parameter Hilbert signed measure convolutions (MP-HSM-C) [26]. All these approaches produce a vector and use XGBoost [15] to train a classifier.

The results in Table 1 indicate that graphcodes are competitive on most of these datasets in terms of accuracy. In terms of runtime performance, the instances are rather small and all approaches terminate within a few seconds (with the exception of GRIL that took longer). Also, while the numbers in Table 1 for the previous approaches are taken from [26], we have partially rerun the classification using convolutional neural networks instead of XGBoost. Since the results were comparable, we decided to use the numbers from the previous work.

Graphcodes do not outperform other methods on these datasets but one can observe that there is no descriptor that consistently outperforms the other descriptors. We also observe that the performance of a certain descriptor on a certain dataset seems a little bit arbitrary. For example, (MP-HSM-C) has arguably the best overall performance but has the worst performance on COX2. A possible explanation could be that there is not enough topological signal in these datasets. This might be unfavorable for graphcodes as they capture more information at the cost of invariance. We also note that the different formats of the topological descriptors require different classifiers and make a direct comparison of the results difficult. This test was included primarily because it is the standard test in related work. Still, it seems unclear that topological descriptors are well suited for these datasets as, for example, on the PROTEINS dataset GNN-architectures reach up to $85\%$ accuracy [38].

Table 1: Graph classification results. The table shows average test set prediction accuracy in %. The numbers in all columns except the last one are taken from Table 3 in [26].

| Dataset | MP-I | MP-K | MP-L | GRIL | MP-HSM-C | GC |
|---------|------|------|------|------|----------|-----|
| PROTEINS | 67.3±3.5 | 67.5±3.1 | 65.8±3.3 | 70.9±3.1 | **74.6±2.1** | 73.6±2.6 |
| DHFR | 80.2±2.2 | 81.7±1.9 | 79.5±2.3 | 77.6±2.5 | **81.9±2.5** | 76.4±3.9 |
| COX2 | 77.9±2.7 | **79.9±1.8** | 79.0±3.3 | 79.8±2.9 | 77.1±3.0 | 78.7±4.9 |
| MUTAG | 85.6±7.3 | 86.1±5.2 | 84.0±6.8 | **87.8±4.2** | 85.6±5.3 | 86.4±6.1 |
| IMBD-BINARY | 71.1±2.1 | 68.2±1.2 | 71.2±2.0 | 65.2±2.6 | **74.8±2.5** | 65.4±2.7 |

Table 2: Average test set prediction accuracy in % over 20 train/test runs with random $80/20$ train/test split on the point cloud dataset and computation time in seconds of the topological descriptors. We note that GRIL could only be computed with low resolution.

| | MP-I | MP-L | P-I | GRIL | MP-HSM-C | GC | GC-NE |
|--|------|------|-----|------|----------|-----|-------|
| Accuracy | 64.1±4.7 | 37.2±1.5 | 43.6±2.2 | 74.9±2.7 | 57.0±2.3 | **86.9±1.4** | 82.8±1.9 |
| Time | 9176 | 3519 | 1090 | 333187 | 282 | **95** | – |

**Shape dataset.** To demonstrate that graphcodes are powerful topological descriptors, we apply them on a synthetic shape dataset with a strong topological signal. We construct 5 classes of shapes $c_0, \ldots, c_4$ as follows: Class $c_i$ consists of $i$ annuli and $5 - i$ disks in the plane. The centers and radii are sampled uniformly such that the shapes do not overlap. This implies that the homology of class $c_i$ in degree one has rank $i$. Now we uniformly sample points from these shapes and add uniform noise. The Figure on the left shows an example of class $c_3$.

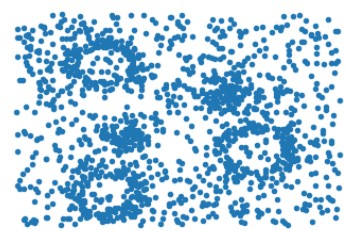 We generate 1000 random shape configurations and point samples per class to obtain a dataset of 5000 point clouds. The point clouds are labeled with the homology of the underlying shape configuration. The goal is to classify the point clouds according to their homology in degree one. To filter the homology signal from the noise, we first compute a local density estimate at each point in a point cloud and compute the Delaunay-bifiltration [2] with respect to relative density scores. This yields a dataset of 5000 labeled bifiltrations. From these bifiltrations, we compute the graphcodes as well as the topological descriptors for the same related approaches as for the graph case. Additionally we also compute one-parameter persistence images (P-I) based on a one-parameter alpha filtration.

The time to compute these topological descriptors is reported in Table 2. Graphcode is faster than every other method, in some cases by orders of magnitude. On the other hand, training graph neural networks is more time-consuming than for convolutional neural networks, and thus graphcodes require more time in the subsequent training phase. In our experiments, the training took around 9 minutes for graphcodes and around 1 minute for other methods.

We now split the various datasets of topological descriptors and class labels $80/20$ into a training set and a test set without labels, train neural networks on the training sets and test their ability to make predictions on the test sets. Further details on these experiments can be found in Appendix C.2 and the supplementary materials. The results in Table 2 show that on this inherently topological classification task, graphcodes outperform every other method by a significant margin. To demonstrate that the graphcode edges that connect consecutive layers add significant information, we run the same experiment on graphcodes with edges removed (GC-NE). The results show even without edges, graphcodes yield a better accuracy compared to related approaches, but also that the edge information further improves accuracy.

**Random point process dataset.** The following is a variation of an experiment proposed in [6]: they consider 4 types of random point processes, namely a Poisson, Matérn, Strauss and Baddeley-Silverman process, and try to discriminate the Poisson null model from the other processes using a hypothesis test based on multiparameter persistent Betti numbers. The latter 3 processes are prototypical models for attractive behaviour, repulsive behaviour and complex interactions, respectively. We instead use multiparameter topological descriptors and neural networks to classify these processes.

Table 3: Average test set prediction accuracy in % over 20 train/test runs with random $80/20$ train/test split on the point-process dataset.

| Dataset | MP-I | MP-L | P-I | GRIL | MP-HSM-C | GC | GC-NE |
|---------|------|------|-----|------|----------|-----|-------|
| Processes | 66.0±2.5 | 50.2±3.0 | 35.5±10.4 | 61.1±1.6 | 70.7±4.9 | **83.4±2.5** | 83.1±3.7 |

Table 4: Orbit classification results. The table shows average test set prediction accuracy in %. The numbers in all columns except the last two are taken from Table 1 in [11].

| Dataset | PSS-K | PWG-K | SW-K | PF-K | PersLAY | GC | GC-NE |
|---------|-------|-------|------|------|---------|-----|-------|
| Orbit5k | 72.4±2.4 | 76.6±0.7 | 83.6±0.9 | 85.9±0.8 | 87.7±1.0 | **88.5±1.1** | 88.4±1.5 |
| Orbit100k | - | - | - | - | 89.2±0.3 | **92.3±0.3** | 91.5±0.3 |

We create a dataset *Processes* consisting of 4 classes, each of which consisting of 1000 point clouds sampled from the above processes and use the topological descriptors and neural networks, discussed for the shape dataset above, for the classification. More details can be found in C.3. The results reported in Table 3 show again that graphcodes outperform other topological descriptors. They also show that for these random point processes the influence of the edges of the graphcodes is much smaller than for the shape dataset. This is expected since prominent persistent features along the density direction are very unlikely in random point processes.

**Orbit dataset.** Finally we test our pipeline on another dataset which has been established in topological data analysis as a benchmark in the one-parameter setting. The purpose of this experiment is twofold: On the one hand it demonstrates that grapohcodes can be applied to very big datasets. On the other hand it compares the two-parameter graphcode pipeline to its one-parameter analog PersLay [11]. The dataset consists of orbits generated by a dynamical system defined by the following rule:

$$\begin{cases} x_{n+1} = x_n + ry_n(1 - y_n) & \mod 1 \\ y_{n+1} = y_n + rx_{n+1}(1 - x_{n+1}) & \mod 1 \end{cases} \qquad (3)$$

where the starting point $(x_0, y_0)$ is sampled uniformly in $[0, 1]^2$. The behaviour of this dynamical system heavily depends on the parameter $r > 0$. Following [11], we create two datasets consisting of 5 classes of orbits of 1000 points generated by this dynamical system, where the 5 classes correspond to the following five choices of the parameter $r = 2.5, 3.5, 4.0, 4.1$ and $4.3$. The datasets *Orbit5k* and *Orbit100k* consist of 1000 and 20000 orbits per class, respectively. We again use our graphcode pipeline, discussed for the shape dataset, to classify them. The computation of the graphcodes of the 100000 point clouds took just 27 minutes demonstrating the efficiency of our algorithm. The results are reported in Table 4 where we compare them to the results achieved by Persistence Scale Space Kernel (PSS-K) [32], Persistence Weighted Gaussian Kernel (PWG-K) [24], Sliced Wasserstein Kernel (SW-K) [12], Persistence Fisher Kernel (PF-K) [25] and (PersLAY) [11] as reported in Table 1 of [11]. The results demonstrate that graphcodes perform better than the one-parameter methods and underpin our conjecture that the performance of the graphcode-GNN pipeline relative to other methods gets better as the size of the dataset increases. More details can be found in C.4.

**Graphcodes with different bases.** The edges of the graphcode of a two-parameter persistence module depend on a choice of bases. So, in all experiments, we used graphcodes with bases produced by our graphcode-algorithm which is based on the standard reduction algorithm. We next discuss the performance of graphcodes on the shape classification task introduced above with respect to different choices of bases. We compute the graphcodes using the dataset from the previous paragraph "Shape dataset". At first we construct a graphcode dataset (GC-ER) using an exhaustive column reduction [4] instead of the standard reduction. Next we construct a graphcode dataset (GC-RS) where we randomly shuffle the bases. This is done by performing valid column additions on the input presentation with a $5\%$ probability. Finally, we produce a graphcode dataset (GC-BC) containing 20 graphcodes constructed with random base shuffles for each input instance.

We observe that the bases chosen by the standard reduction and exhaustive reduction algorithm are far from random. The input presentation arising from a simplicial bifiltration is usually sparse. We find that this initial sparseness is preserved by the standard and exhaustive reduction algorithm in the sense that both lead to sparse graphcodes. If we do random base changes in the input presentation we reduce the sparseness of the input which also leads to a loss of sparseness in the output graphcodes.

Table 5: Average test set prediction accuracy in % over 20 train/test runs of Graphcodes with different choices of bases on the shape dataset.

|  | GC | GC-NE | GC-ER | GC-RS | GC-BC |
|---|---|---|---|---|---|
| 100 Epochs | **86.9±1.4** | 82.8±1.9 | 86.7±1.4 | 84.5±2.4 | 86.6±1.7 |
| 200 Epochs | 87.1±1.6 | 83.7±1.0 | 87.0±1.4 | 85.0±1.7 | **88.1±1.2** |

The result is an increase in the average number of edges of the produced graphs. Average number of edges with standard reduction: $\sim 826$, exhaustive reduction: $\sim 841$, random shuffle: $\sim 1977$.

We train the same graph neural network as in the previous experiments on these alternative graphcode datasets and report the results in Table 5. For the (GC-BC) dataset we modify the training process in the following way: In the $i$-th epoch of the training process we use the $(i \bmod 20)$-th graphcode for each instance. This approximates a change of basis of each graphcode after each training epoch by picking one of 20 available bases. We find that this training procedure disproportionally benefits from a larger number of training epochs. Therefore, we run the same experiments with twice the number of training epochs.

The results in Table 5 show that the exhaustive reduction (GC-ER) does not significantly change the result compared to the standard bases (GC). The random basis shuffle (GC-RS) leads to slightly worse performance and a slight increase in variability of the results but we note that the performance is still better than without edges (GC-NE). If we use randomly shuffled bases but provide the network 20 different bases for each instance we match, and with more training epochs, even exceed the performance of (GC) and (GC-ER). These results indicate that changing the graphcode bases during the training process can increase the performance.

## 7   Conclusion

Our shape experiment shows that current implementations of topological classifiers struggle with simple datasets that contain a clear topological signal but also a lot of noise. A possible explanation is that the vectorization step in these methods blurs the features too much or relies on invariants which might be too weak for a classifier to pick up delicate details. Graphcodes, on the other hand, provide an essentially complete description of the (persistent) topological properties of the data and delegates finding the relevant signal to the graph neural network. As additional benefit, some vectorizations are challenging to compute, whereas graphcodes can be computed efficiently.

The biggest drawback of graphcodes is certainly that they dependent on a choice of basis and therefore are not uniquely defined for a given dataset. The bases chosen by the standard reduction algorithm are special in the sense that they lead to sparse graphs. Doing random basis changes on the graphcode dataset leads to denser graphs and a slightly worse performance on the shape classification task. But doing multiple random basis changes during the training process and, thus, providing the neural network different bases for the same instance, increases the performance even beyond the performance of the sparse graphs.

A goal for future work is to combine sparse graphcodes and random basis changes during the training process. A possible direction could be to decompose the two-parameter modules into indecomposable summands, compute the graphcodes for the indecomposables and perform random basis changes on the individual components during the training process. By working only with graphcodes of decompositions we could reduce the number of possible edges.

We speculate that the combination of computational efficiency and discriminatory power will make graphcodes a valuable tool in data analysis. With the advent of more efficient techniques to generate bifiltrations for large datasets, we foresee that the potential of graphcodes will be a study of investigation in the coming years.

## Acknowledgments and Disclosure of Funding

This research has been supported by the Austrian Science Fund (FWF), grant numbers W1230 and P 33765-N.

The authors thank David Loiseaux for his help with using the *multipers*-package and Shreyas Samaga and Soham Mukherjee for their help with using the *GRIL*-package.

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

# A  Basic topological notions

An *(abstract) simplicial complex* $K$ with vertex set $V$ is a collection of subsets of $V$, called *simplices*, with the property that whenever $\sigma \in K$ and $\tau \subseteq \sigma$, then $\tau \in K$ as well. In that case, $\tau$ is called a *face* of $\sigma$. A *subcomplex* $L$ of $K$ is a subset of $K$ that is itself a simplicial complex. The dimension of a simplex is its number of vertices minus $1$ – this corresponds to the interpretation that the vertices are embedded in Euclidean space and a simplex is identified with the convex hull of the embedded vertices that define the simplex. Simplices in dimension 0, 1, 2 are called vertices, edges, and triangles, respectively. A *facet* of a $p$-simplex $\sigma$ is a face of $\sigma$ that has dimension $p-1$. A $p$-simplex has exactly $(p+1)$-facets.

Let $\mathbb{Z}_2$ the field with two elements, $p \geq 0$ an integer and $K$ be a simplicial complex. The set of $p$-simplices forms the basis of a $\mathbb{Z}_2$-vector space $C_p(K)$, and we call elements of this vector space *p-chains*. Put differently, a chain is a formal linear combination of $p$-simplices with coefficients in $\{0, 1\}$ and can thus be interpreted as a subset of $p$-simplices. The *boundary* of a $p$-simplex $\sigma$, written $\partial \sigma$, is the $(p-1)$-chain formed by all facets of $\sigma$. For instance, the boundary of a triangle is the sum of the three edges that (geometrically) form the boundary of the triangle. The boundary extends to a linear map $\partial : C_p(K) \to C_{p-1}(K)$ because it is defined on a basis of $C_p(K)$.

A $p$-chain $c$ is called a *p-cycle* if $\partial c = 0$. The $p$-cycles are the kernel elements of the map $\partial$ and hence form a vector space which we denote by $Z_p(K)$. The aforementioned three edges bounding a triangle form a 1-cycle because when taking their boundary, every vertex appears twice, and hence vanishes because we work with $\mathbb{Z}_2$-coefficients.

A $p$-chain $c$ is called a *p-boundary* if there exists a $(p+1)$-chain $d$ such that $\partial d = c$. In other words, the $p$-boundaries are the images of the map $\partial$ and hence, form a vector space that we denote by $B_p(K)$. A fundamental property is that for any chain $c$, we have that $\partial(\partial c) = 0$. As an example, for $c$ a triangle, the above examples exemplify that indeed, the boundary of the boundary of the triangle is trivial. As a consequence, $p$-boundaries are $p$-cycles and hence $B_p(K)$ is a subspace of $Z_p(K)$.

The *p-th homology group* $H_p(K)$ is the quotient $Z_p(K)/B_p(K)$. We will use the standard notion of "homology group" even though it is even a vector space in our case. The rank of $H_p(K)$ can be interpreted as the number of $p$-dimensional holes in $K$: indeed, a "hole" in $K$ must have a $p$-cycle (i.e., an element of $Z_p(K)$) that encloses the hole, but we need to disregard such cycles that loop around a space that is filled by $(p+1)$-simplices, hence we divide out $B_p(K)$. We write $[\alpha]_K$ for the elements of $H_p(K)$, called *homology classes*, where $\alpha$ is a $p$-cycle, called a *representative cycle* of $[\alpha]_K$. By definition of quotients, $[\alpha]_K = [\alpha']_K$ if and only if $\alpha + \alpha' \in B_p(K)$. When the complex $K$ is clear from context, we also just write $[\alpha]$.

# B  Computation

**Computing barcode bases.**   We start by reviewing the classical matrix reduction algorithm for computing persistent homology. For a fixed filtration of a simplicial complex $K$ and $p \geq 0$, we put the $(p+1)$-simplicies of $K$ in a total order that respects the filtration, meaning that if a $(p+1)$-simplex $\sigma$ of $K$ enters the filtration at index $i$ and a $(p+1)$-simplex $\tau$ of $K$ enters at index $j$ with $i < j$, then $\sigma$ precedes $\tau$ in the order. Writing $n_{p+1}$ for the number of $(p+1)$-simplices of $K$, we can assign to each $(p+1)$-simplex an index in $\{1, \ldots, n_{p+1}\}$ that reflects this order. Doing the same for the $p$-simplices of $K$ (with total number $n_p$), the *boundary matrix* $\partial$ of $K$ in dimension $p$ is defined as the $(n_p \times n_{p+1})$-matrix over $\mathbb{Z}_2$ where each row corresponds to a $p$-simplex and each column to a $(p+1)$-simplex, and the entry at position $(i, j)$ equals 1 if and only if the $i$-th $p$-simplex is a facet of the $j$-th $(p+1)$-simplex.

The columns of $\partial$ span $B_p(K)$. Recall our simplifying assumption that $H_p(K)$ is trivial, implying that the columns also span $Z_p(K)$. However, they are not necessarily a basis because of linear dependance. To obtain a basis, we apply *matrix reduction* on $\partial$: for a non-zero column, denote by its *pivot* the highest index whose coefficient is not zero. Then, traverse the columns from left to right, writing $c_j$ for the currently considered column. As long as $c_j$ is not zero and has the same pivot as a previous column $c_i$ with $i < j$, add $c_i$ to $c_j$. Every column addition will make the pivot entry in $c_j$ disappear since we work with $\mathbb{Z}_2$-coefficients.

This process results in a *reduced matrix* $R$ in which no two columns have the same pivot. The remaining non-zero columns of $R$ are linearly independent and thus form a basis of $Z_p(K)$. The bar attached to each cycle can easily be read off the matrix $R$: fixing a cycle $\alpha$ represented by column $j$, let $\sigma$ be the $(p+1)$-simplex that corresponds to column $j$ in $\partial$. Then the death index is the minimal $d$ such that $\sigma$ is contained in $K_d$. Furthermore, let $i$ denote the pivot of column $j$ and let $\tau$ be the $p$-simplex that corresponds to row $i$ in $\partial$. Then, the birth index is the smallest $b$ such that $\tau$ is contained in $K_b$. Exploiting the fact that the matrix reduction only performs left-to-right column additions and that the resulting basis has pairwise-disjoint pivots, one can show that this basis forms a barcode basis of $Z_p(K)$.

Moreover, the reduction process has the additional property that any column that is ever added to another column is already reduced, that is, will not be modified further in the process. We record this for later:

**Proposition B.1.** *If a matrix $A$ gets reduced to $R$ as above, and a column addition $c_j \leftarrow c_i + c_j$ happens during the reduction process, then $c_i$ is a column of $R$.*

Being a variant of Gaussian elimination, the algorithm to compute a barcode basis out of the boundary matrix $\partial$ requires $O(n^3)$ in the worst case (with $n = \max\{n_{p+1}, n_p\}$). However, the initial sparseness of $\partial$ results in a close-to-linear performance in many applied scenarios [5, 29].

The assumption that $H_p(K)$ is trivial might seem crucial for the construction, but one can lift this assumption without much computational overhead. In that case, one also needs to perform matrix reduction on the boundary matrix spanned by $(p-1)$- and $p$-simplices and do some additional book-keeping to obtain basis elements for $p$-cycles that do not die in $K$. We omit details for brevity.

**Efficient graphcodes through batched matrix reduction.** We consider the algorithmic problem to compute the graphcode of a bifiltration of a simplicial complex $K$. We assume that the bifiltration is 1-*critical* which means that for every simplex $\sigma$, there is a index pair $(i_0, j_0)$, such that $\sigma \in K_{i,j}$ if and only if $i_0 \leq i$ and $j_0 \leq j$. In other words, every simplex has a unique entrance time into the bifiltration; while this assumption does not apply for every bifiltration, it is still satisfied for many instances that occur in practice; furthermore, there are techniques to transform other types of bifiltrations to the 1-critical case [13].

Our input is a list of simplices of $K$ together with a critical index pair $(i_0, j_0)$ per simplex, defining a 1-critical bifiltration as in (2). The output is the graphcode of the bifiltration.

The straight-forward approach to compute the graphcode is to first compute a barcode basis for each horizontal slice $\ell$ independently, obtaining the vertices of the graphcode. Then, express every basis element at level $\ell$ using the barcode basis at level $\ell + 1$ to determine the edges between levels $\ell$ and $\ell + 1$. This requires to solve one linear system per basis element. With $s$ the total number of horizontal slices and $n$ the number of simplices, this approach requires $O(sn^3)$ to compute the barcodes bases and also $O(sn^3)$ to get the edges, since every linear system can be solved in $O(n^2)$ time using the reduced matrices.

The straight-forward approach is non-optimal because it computes the barcode bases on each level from scratch. Since $K_{\ell+1}$ contains $K_\ell$, we can devise a more efficient strategy to update a barcode basis for $K_\ell$ to a barcode basis for $K_{\ell+1}$. In this way, we obtain the barcode bases for all horizontal filtrations in $O(n^3)$ time.

First, we sort all $(p+1)$-simplices of $K$ with respect to their second critical index, and refine to a total order, assigning to each $(p+1)$-simplex an integer in $\{1, \dots, n_{p+1}\}$. Initialize $A$ to be an empty matrix, whose number of columns equals $n_{p+1}$. Precompute for every level $\ell$ the set of $(p+1)$-simplices that are contained in $K_\ell$, but not in $K_{\ell-1}$ (these are precisely those simplices whose first critical value equals $\ell$), calling them the $\ell$-*th batch*. Now assume that $A$ contains a barcode basis for the horizontal level $\ell - 1$. Add the columns of the $\ell$-th batch to $A$ at the appropriate place with respect to the chosen total order and apply the matrix reduction from the previous paragraph on $A$. The resulting reduced matrix yields a barcode basis for level $\ell$.

This above algorithm computes all vertices of the graphcode in worst-case $O(n^3)$ time, and also efficiently in practice, as it basically performs a single reduction of the boundary matrix of $K$, in some order that is determined by the bifiltration. What is perhaps remarkable is that with some extra bookkeeping, the algorithm also computes the edges of the graphcode on the fly. To see that, consider a non-zero column of the boundary matrix before the $\ell$-th batch gets added, representing a basis

element $\alpha$ of $Z_p(K_{\ell-1})$. If the column does not change during the reductions caused by the batch, $\alpha$ is also basis element in the barcode basis for $K_\ell$, and there is a single edge connecting the two copies of $\alpha$ in the basis for $K_{\ell-1}$ and $K_\ell$. If the column changes, this is caused by a column addition with a column from the left. Because of Proposition B.1 the added column is a basis element $z_1$ of the $\ell$-th barcode basis, and we know that $\alpha = z_1 + \alpha^{(1)}$, where $\alpha^{(1)}$ is the cycle represented by the column after the addition. Now if $\alpha^{(1)}$ gets modified by another basis element $z_2$ of the $\ell$-th barcode basis, we get $\alpha = z_1 + z_2 + \alpha^{(2)}$, and so on. The process either stops when the column becomes $0$, in which case $\alpha^{(k)} = 0$ for some $k$, and we have obtained the linear combination of $\alpha$, or the process stops because some $\alpha^{(k)}$ is reduced and will not be modified further in the reduction. In that case, $\alpha^{(k)}$ is itself a basis element $z_{k+1}$ of the $\ell$-th barcode basis, and we obtain that $\alpha = z_1 + z_2 + \ldots + z_{k+1}$. Storing the linear combination during the process does not affect the running time, hence the graphcode can be computed in $O(n^3)$ time.

**Speed-ups.** When implementing the above algorithm, we observed that the resulting graphcodes can become rather large and the practical bottleneck in the computation is to merely create the graphcode data structure. We suggest two ways to reduce the size, resulting in a much better performance: first of all, standard constructions for bifiltrations result in a large number of horizontal slices to consider. Instead, we propose to fix an integer $s > 0$, and to equidistantly split the parameter range of the first critical value equidistantly at $s$ positions, obtaining $s$ slices in the graphcode. To reduce the size of individual graphcodes, we propose to only consider *relevant* bars, where relevance means that the persistence of the bar (i.e., the distance between death and birth) is above some threshold. In particular, this removes bars of zero length from consideration, which are often the majority of all bars in a barcode. We then only return the subgraph of the graphcode induced by the relevant bars. In Figure 3 (right), we have applied this filter, for instance.

Finally, we observed significant performance gains by first computing a *minimal presentation* [20] of the input bifiltration, and computing the graphcode of this minimal presentation. A minimal presentation consists of generators and relations capturing the homology of a bifiltration. The entire approach works in an analogous way, with generators taking the role of $p$-simplices and relations the role of $(p + 1)$-simplices. We skip details for brevity.

## C    Details on experiments

In all our experiments, we compute the graphcodes from minimal presentations using our C++ library. The graphcode software requires parameters specifying the homology degree, the number of slices and the direction in which to take the slices. If the minimal presentation is already computed for a specific homology degree (an option provided by MPFREE bitbucket.org/mkerber/mpfree and FUNCTION_DELAUNAY bitbucket.org/mkerber/function_delaunay), the degree parameter can be omitted. One can also specify a relevance threshold $t$ to obtain the subgraph induced by all nodes $(b, d)$ such that $d - b > t$.

### C.1    Graph experiments

For the datasets listed in Table 1 we first compute the Heat Kernel Signature-Ricci Curvature bifiltration using a function provided by github.com/TDA-Jyamiti/GRIL and then compute a minimal presentation of this filtration using the software MPFREE [20]. In all the graph experiments we compute the graphcodes using homology degree one, 20 slices and relevance-threshold $t = 0$. We tested both possible slicing directions controlled by the option "primary-parameter" in the graphcode software and found that primary parameter 1 is slightly better. The computation of the graphcodes of these datasets takes between 1 and 7 seconds. We additionally augment the vertices $[(b, d), i]$ of the raw graphcodes with their multiplicative $\frac{d}{b}$ and additive $d - b$ persistence. The vertex attributes $[(b, d, \frac{d}{b}, d - b), i]$ yield slightly better results. The slices index $i$ is only used for the local max-pooling and not as part of the vector attributes for the neural network.

We then train a graph neural network classifier on these graphcode datasets where we use the architecture specified in Section 5. The details of the chosen parameters for the GNN architectures can be found in the code provided as supplementary material. We randomly shuffle the dataset and split it $80/20$ into a labeled training set and a test set without labels, train the GNN on the training

set and evaluate it on the test set. We run this procedure 20 times and average the achieved test set prediction accuracy over this 20 train/test runs. The results are reported in Table 1.

## C.2 Shape experiments

The point cloud dataset is generated as follows. For class $i$ we have to put $i$ annuli and $5 - i$ disks on an empty canvas in a way such that they don't overlap. We put the shapes on the canvas one by one by uniformly sampling radii and centers in such a way that a newly added shape would have at least separation $\epsilon > 0$ from all shapes that are already there. At the end we put uniform noise with a uniformly sampled density over the whole canvas. For each of the 5 classes we generate 1000 of these random shape configurations and take a point sample from each of them. This leads to a dataset of 5000 point clouds in the plane. The details of all the chosen parameters can be found in the code in the supplementary materials.

The point clouds have a lot of randomness to them. The only thing that two point clouds in the same class have in common is that, before adding the noise, they are both sampled from a space with the same homology in degree one. A human could probably still predict with high accuracy how many annuli are in a picture (cf. the example figure in Section 6). This is because the underlying regions of the shapes have much higher density. To enable a classifier based on topological descriptors to do a similar kind of inference we have to introduce some measure of density. Hence, we compute a local density estimate at every point of a point cloud based on the number of neighbours in a circle with a given radius. We then score the points with respect to these local density estimates and use these score values as function values for the Delaunay-bifiltration computed with FUNCTION_DELAUNAY [2].

For the graphcode computation we use homology degree one, 10 slices and we slices the bifiltration in direction of fixed density. On these datasets we have to set a positive relevance-threshold of $0.1$ because the resulting graphcodes would be too big for the available GPU memory in the GNN training process. The density scores and the slicing along fixed density will yield the following slices: The first slice contains those 5% of the points with the lowest density. The second slice contains those 10% of the points with the lowest density, etc. The computation time reported in Table 2 is the time needed to compute the graphcodes of the whole dataset from the minimal presentations.

We then train a graph neural network classifier on these graphcode datasets where we use the architecture specified in Section 5. The details of the chosen parameters for the GNN architectures can be found in the code provided as supplementary material. We randomly shuffle the dataset and split it $80/20$ into a labeled training set and a test set without labels, train the GNN on the training set and evaluate it on the test set. We run this procedure 20 times and average the achieved test set prediction accuracy over this 20 train/test runs. We repeat this experiment for the graphcodes after removing all the edges. The results are reported in Table 2.

As a comparison we test various other topological descriptors on this classification task. We start with one-parameter persistence images obtained from one-parameter alpha-filtrations in homology degree one on the point clouds. For the computation we use the Gudhi package [33]. The computation time reported in Table 2 is the time needed to compute the persistence images of the whole dataset from the alpha filtrations.

Next we compute the multiparameter persistence images, landscapes and the signed measure convolutions using the multipers package github.com/DavidLapous/multipers. These vectorizations are computed from minimal presentations, for homology degree one, of the Delaunay-bifiltrations. For all these vectorizations we use a resolution of $100 \times 100$, i.e., the output are $100 \times 100$ images. For the landscapes we use the first 5 landscapes. The computation time reported in Table 2 is the time needed to compute the vectorizations of the whole dataset from the minimal presentations.

Finally we compute the generalized rank invariant landscapes (GRILs), for homology degree one, using the GRIL package github.com/TDA-Jyamiti/GRIL/. Since the GRIL package takes bifiltrations as input we first have to compute the Delaunay-bifiltrations in non-presentation form and convert them to inputs suitable for GRIL. Since the computation of the GRIL's is costly we were forced to choose a rather large step size to make the computation of the landscapes feasible. We found that enlarging the step size lead to better results than reducing the resolution. The resulting images are of size $17 \times 17$. The computation time reported in Table 2 is the time needed to compute the landscapes of the whole dataset from the precomputed GRIL inputs.

For the computation of all multiparameter vectorizations we first scale the bidegrees of the bifiltration, i.e., the parameter of the alpha complex and the density scores, to one to make the two parameters comparable.

We then train convolutional neural network classifiers on these image datasets. As in the GNN case, we randomly split the datasets into training and test sets using a $80/20$ split, train the network on the training set and test it on the test set. We run this procedure 20 times and take the average test set prediction accuracy. The results are reported in Table 2.

We note that despite the big step size in the GRILs, leading to rather coarse images, the performance is quite good compared to other vectorization methods. We believe that, given the computational resources to use a smaller step size, the GRILs would perform significantly better.

## C.3 Random Point-Process Experiments

Following [6], we consider four classes of point processes and create the dataset *Processes* by simulating random samples of these processes. The four classes of our dataset correspond to the four different processes and consist of 1000 random samples per class. We simulate all processes in two-dimensional Euclidian space where we restrict the sampling window to $[0, 1]^2$. The first process is a standard homogeneous Poisson process which, in our case, corresponds to uniformly sampling a Poisson distributed number of points with a given intensity in $[0, 1]^2$. The second process is a Matérn cluster process which is based on a parent Poisson process, whose points can be viewed as cluster centers, where each parent point creates a Poisson distributed number of child points uniformly sampled in a sphere centered at the parent. The third process is a Strauss process which models repulsive behaviour. In the Strauss process there is a penalty on points sampled within a given distance of each other based on an interaction parameter. To sample these processes, we use the functions PoissonPointProcess, MatérnPointProcess and StraussPointProcess in MATHEMATICA. The last process we consider is a Baddeley-Silverman process where we subdivide the sampling window $[0, 1]^2$ into a grid of boxes and sample 0, 1 or 2 points in each box with probabilities 0.45, 0.1 and 0.45, respectively. Since we could not find an implementation of this process in MATHEMATICA we implemented this process in PYTHON. The details of all parameter choices can be found in the supplementary materials. We choose the parameters in such a way that a sample of any of the above processes contains about 200 points on average.

As in the previous experiments we compute a Delaunay-bifiltration based on local density estimates and then compute graphcodes in homology degree one using 10 slices along fixed density values without a threshold. The results are reported in Table 4.

We then train a graph neural network classifier on this graphcode dataset where we use the architecture specified in Section 5. The details of the chosen parameters for the GNN architectures can be found in the code provided as supplementary material. We randomly shuffle the dataset and split it $80/20$ into a labeled training set and a test set without labels, train the GNN on the training set and evaluate it on the test set. We run this procedure 20 times and average the achieved test set prediction accuracy over this 20 train/test runs. We repeat this experiment for the graphcodes after removing all the edges. The results are reported in Table 3.

As a comparison we also compute persistence images based on a one-parameter alpha filtration and multiparameter persistence images, landsacapes and signed measure convolutions as well as generalized rank invariant landscapes from the bifiltrations and classify them using convolutional neural networks. For all these experiments we use the same settings as for the shape datasets. The results can be found in Table 3.

## C.4 Orbit Experiments

The orbit datasets *Orbit5k* and *Orbit100k* are created as follows. For each of the five parameter values $r = 2.5, 3.5, 4.0, 4.1$ and $4.3$ we uniformly sample 1000 and 20000 points $(x_0, y_0)$ in $[0, 1]^2$, respectively, and run the dynamical system (3) for 1000 steps. In this way we obtain the two datasets *Orbit5k* and *Orbit100k* consisting of 5 classes of 1000 and 20000 point clouds, respectively, where each point cloud consists of 1000 points in $\mathbb{R}^2$. The class labels are the values of $r$ used to generate a point cloud.

After constructing the datasets we compute local density estimates at every point, score the points with respect to these density estimates and compute a Delaunay-bifiltration with respect to these density scores. In contrast to the point clouds of the shape dataset, the point clouds from the orbit datasets do not have particularly prominent dense regions which explains why the the difference between the methods based on one-parameter persistence and graphcodes is smaller than for the shape dataset. From these bifiltrations we compute graphcodes in homology degree one, using 10 slices along fixed density values and use a persistence threshold of 0.002.

We then train a graph neural network classifier on these graphcode datasets where we use the architecture specified in Section 5. The details of the chosen parameters for the GNN architectures can be found in the code provided as supplementary material. We randomly shuffle the dataset and split it 70/30 (to be consistent with [11]) into a labeled training set and a test set without labels, train the GNN on the training set and evaluate it on the test set. We run this procedure 20 times for *Orbit5k* and 10 times for *Orbit100k* and average the achieved test set prediction accuracy over this train/test runs. We note that in [11] they average over 100 train/test runs. For time reasons, especially on the relatively large *Orbit100k* dataset, we avoided such a large number of runs but, since the results have low variability, this does not make a significant difference. We repeat this experiment for the graphcodes after removing all the edges. The results are reported in Table 4. We note that in [11] they use persistent homology in degree zero and degree one for the classification. Thus we achieve the reported accuracy with less information. We can observe that the performance of graphcodes relative to perslay and the other methods as well as the influence of the graphcode-edges increases as the size of the dataset increases. This demonstrates that the true power of the combination of graphcodes and graph neural networks really starts to manifest itself on larger datasets.

## D  Graphcodes of general two-parameter persistence modules

In Section 3, we defined graphcodes of two-parameter persistence modules arising from bifiltered simplicial complexes. In this section, we show that graphcodes can be defined for arbitrary two-parameter persistence modules. For the graphcode construction, we consider a two-parameter persistence module as a sequence of one-parameter persistence modules connected by morphisms. A one-parameter persistence module $M$ is a diagram

$$M_1 \xrightarrow{M_1^2} M_2 \xrightarrow{M_2^3} \cdots \xrightarrow{M_{n-2}^{n-1}} M_{n-1} \xrightarrow{M_{n-1}^n} M_n$$

where $M_i$ is a finite-dimensional vector space and $M_i^{i+1} \colon M_i \to M_{i+1}$ is a linear map. The elementary building blocks of one-parameter persistence modules are the so-called *interval modules* $\mathbf{I}_{[a,b)}$ defined by

$$(\mathbf{I}_{[a,b)})_i := \begin{cases} \Bbbk & \text{if } a \leq i < b \\ 0 & \text{else} \end{cases}$$

$$(\mathbf{I}_{[a,b)})_i^{i+1} := \begin{cases} \text{id} & \text{if } a \leq i < b-1 \\ 0 & \text{else} \end{cases}$$

A morphism of one-parameter persistence modules $\phi \colon M \to N$ is a collection of linear maps $\phi_i \colon M_i \to N_i$ such that the following diagram commutes:

$$\begin{array}{ccccccccc} N_1 & \xrightarrow{N_1^2} & N_2 & \xrightarrow{N_2^3} & \cdots & \xrightarrow{N_{n-2}^{n-1}} & N_{n-1} & \xrightarrow{N_{n-1}^n} & N_n \\ \phi_1 \uparrow & & \phi_2 \uparrow & & & & \phi_{n-1} \uparrow & & \phi_n \uparrow \\ M_1 & \xrightarrow{M_1^2} & M_2 & \xrightarrow{M_2^3} & \cdots & \xrightarrow{M_{n-2}^{n-1}} & M_{n-1} & \xrightarrow{M_{n-1}^n} & M_n \end{array}$$

The theorems of Krull-Remak-Schmidt [3, Theorem 1] and Gabriel [21, Chapter 2.2] imply that every one-parameter persistence module $M$ is isomorphic to a unique direct sum of interval modules, i.e., $M \cong \bigoplus_{j=1}^g \mathbf{I}_{[a_j,b_j)}$. We define by $\mathrm{Dgm}(M) := \{(a_j, b_j) \in \mathbb{R}^2 | 0 \leq j \leq g\}$ the *persistence diagram* of $M$. The points or intervals $(a_j, b_j) \in \mathrm{Dgm}(M)$ uniquely determine $M$ up to isomorphism. We call an isomorphisms $\mu \colon M \xrightarrow{\cong} \bigoplus_{j=1}^g \mathbf{I}_{[a_j,b_j)}$ a *barcode basis* of $M$. This is the abstract analog of the barcode basis of Definition 2.1. Note that there might be many choices for such an isomorphism.

The results discussed above can be interpreted on an elementary level in the following way: for a persistence module $M$ there exists a choice of bases of the vector spaces $M_i$ such that all the matrices

$M_i^{i+1}$ are in diagonal form, i.e., every basis element in $M_i$ is either mapped to a unique basis element in $M_{i+1}$ or is mapped to zero. Since there is no unique way of transforming arbitrary bases of $M$ into a barcode basis there is no unique isomorphism.

Since every persistence module is isomorphic to a direct sum of interval modules, to understand morphisms of persistence modules, it is enough to understand morphisms between interval modules. Given two interval modules $\mathbf{I}_{[a,b)}$ and $\mathbf{I}_{[c,d)}$ the vector space $\operatorname{Hom}\big(\mathbf{I}_{[a,b)}, \mathbf{I}_{[c,d)}\big)$ of morphisms $\mathbf{I}_{[a,b)} \to \mathbf{I}_{[c,d)}$ has the following simply structure:

$$\operatorname{Hom}\big(\mathbf{I}_{[a,b)}, \mathbf{I}_{[c,d)}\big) \cong \begin{cases} \Bbbk & \text{if } c \leq a < d \leq b \\ 0 & \text{else} \end{cases} \tag{4}$$

This means that, if the intervals overlap as described in (4), then, up to a scalar factor $\lambda \in \Bbbk$, there is a unique morphism $\mathbf{I}_{[a,b)} \xrightarrow{\lambda} \mathbf{I}_{[c,d)}$. Otherwise the only possible morphism is the zero-morphism. For a choice of barcode bases $\mu\colon M \xrightarrow{\cong} \bigoplus_{j=1}^{g} \mathbf{I}_{[a_j,b_j)}$ and $\nu\colon N \xrightarrow{\cong} \bigoplus_{l=1}^{h} \mathbf{I}_{[c_l,d_l)}$, a morphism $\phi\colon M \to N$ induces a morphism $\psi^\phi$

$$\begin{array}{ccc} M & \xrightarrow{\phi} & N \\ \mu \downarrow & & \downarrow \nu \\ \bigoplus_{j=1}^{g} \mathbf{I}_{[a_j,b_j)} & \xrightarrow{\psi^\phi} & \bigoplus_{l=1}^{h} \mathbf{I}_{[c_l,d_l)} \end{array}$$

defined by $\psi^\phi := \nu \circ \phi \circ \mu^{-1}$. Such a morphism between direct sums is completely determined by the morphisms between individual summands, i.e.

$$\operatorname{Hom}(M, N) \cong \bigoplus_{j=1}^{g} \bigoplus_{l=1}^{h} \operatorname{Hom}\big(\mathbf{I}_{[a_j,b_j)}, \mathbf{I}_{[c_l,d_l)}\big)$$

The morphisms between summands are given by composition with the inclusion and projection to these summands

$$\begin{array}{ccc} \mathbf{I}_{[a_s,b_s)} & \xrightarrow{\psi_{ts}^\phi} & \mathbf{I}_{[c_t,d_t)} \\ \iota_s \downarrow & & \uparrow \pi_t \\ \bigoplus_{j=1}^{g} \mathbf{I}_{[a_j,b_j)} & \xrightarrow{\psi^\phi} & \bigoplus_{l=1}^{h} \mathbf{I}_{[c_l,d_l)} \end{array}$$

Hence, by (4), $\psi_{ts}^\phi := \pi_t \circ \psi^\phi \circ \iota_s$ is either zero or determined by a scalar $\lambda_{ts}^\phi \in \Bbbk$ and we can represent the morphism $\psi^\phi$ by a matrix $\mathcal{M}(\phi)$ of the form

$$\begin{array}{c} \\ [c_1,d_1) \\ [c_2,d_2) \\ \vdots \\ [c_h,d_h) \end{array} \begin{array}{c} [a_1,b_1) \quad [a_2,b_2) \quad \cdots \quad [a_g,b_g) \\ \left( \begin{array}{cccc} \lambda_{11}^\phi & \lambda_{12}^\phi & \cdots & \lambda_{1g}^\phi \\ \lambda_{21}^\phi & \lambda_{22}^\phi & \cdots & \lambda_{2g}^\phi \\ \vdots & \vdots & \ddots & \vdots \\ \lambda_{h1}^\phi & \lambda_{h2}^\phi & \cdots & \lambda_{hg}^\phi \end{array} \right) \end{array}$$

where $\lambda_{ts}^\phi$ is the scalar determining the morphism $\psi_{ts}^\phi$. Note the analogy to matrices representing maps between vector spaces with respect to a choice of basis.

*Example* D.1. Consider the following morphism of persistence modules

$$\begin{array}{ccccccccc} N\colon & & \Bbbk & \xrightarrow{\binom{1}{0}} & \Bbbk^2 & \xrightarrow{(1\ 0)} & \Bbbk & \longrightarrow & 0 \\ \phi \uparrow & & (0) \uparrow & & \uparrow \binom{1}{1} & & \uparrow (1) & & \uparrow \\ M\colon & & 0 & \xrightarrow{(0)} & \Bbbk & \xrightarrow{(1)} & \Bbbk & \longrightarrow & 0 \\ & & 1 & & 2 & & 3 & & 4 \end{array}$$

In this case, we have $M = \mathbf{I}_{[2,4)}$ and $N = \mathbf{I}_{[1,4)} \oplus \mathbf{I}_{[2,3)}$, i.e., $M$ and $N$ are already in barcode form. The morphism $\phi \colon M \to N$ given by the vertical maps sends $\mathbf{I}_{[2,4)}$ to both summands $\mathbf{I}_{[1,4)}$ and $\mathbf{I}_{[2,3)}$. Therefore, we obtain

$$
\mathcal{M}(\phi) = \begin{matrix} & [2,4) \\ [1,4) \\ [2,3) \end{matrix} \begin{pmatrix} 1 \\ 1 \end{pmatrix} \tag{5}
$$

As in the case of matrix representations of linear maps, representing $\phi$ with respect to different bases leads to different coefficients. Therefore, the matrix $\mathcal{M}(\phi)$ is not unique. It depends on the choice of barcode bases $\mu$ and $\nu$.

*Example* D.2. The morphisms of persistence modules $\phi_{\text{front}}$ and $\phi_{\text{back}}$ given by the front- and back-face of the following diagram are isomorphic

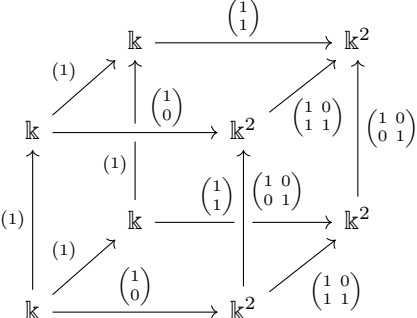

but they induce the following different matrices

$$
\mathcal{M}(\phi_{\text{front}}) = \begin{matrix} & [1,3) \\ [1,3) \\ [1,3) \end{matrix} \begin{pmatrix} 1 \\ 0 \end{pmatrix}
$$

$$
\mathcal{M}(\phi_{\text{back}}) = \begin{matrix} & [1,3) \\ [1,3) \\ [1,3) \end{matrix} \begin{pmatrix} 1 \\ 1 \end{pmatrix}
$$

To get rid of the dependence on scalar factors, from now on we assume that $\mathbb{k} = \mathbb{Z}_2$. This implies that the entries of $\mathcal{M}(\phi)$ are either 0 or 1 and allows us to view

$$
\begin{pmatrix} 0 & \mathcal{M}(\phi)^T \\ \mathcal{M}(\phi) & 0 \end{pmatrix}
$$

as the adjacency-matrix of a bipartite graph with vertex set $\operatorname{Dgm}(M) \cup \operatorname{Dgm}(N)$.

**Definition D.3** (Graphcode general). A graphcode $\mathcal{G}(\phi) = \big(V(\phi), E(\phi)\big)$ of a morphism $\phi \colon M \to N$ of one-parameter persistence modules with respect to a choice of barcode bases is the bipartite graph defined by

$$
\begin{aligned}
V(\phi) &:= \operatorname{Dgm}(M) \cup \operatorname{Dgm}(N) \\
E(\phi) &:= \{(v,w) \in \operatorname{Dgm}(M) \times \operatorname{Dgm}(N) \mid \mathcal{M}(\phi)_{wv} = 1\}
\end{aligned}
$$

By construction, the graphcode describes the morphism $\phi \colon M \to N$ up to isomorphism. In some sense a graphcode is just a representation of the morphism with respect to specific (barcode) bases.

We can now extend graphcodes from a single morphism of one-parameter persistence modules to two-parameter persistence modules which can be viewed as a sequence of one-parameter persistence

modules:

$$
\begin{array}{ccccccc}
M_{\bullet m} & M_{0m} & \xrightarrow{M_{0m}^{1m}} & M_{1m} & \xrightarrow{M_{1m}^{2m}} & \cdots & \xrightarrow{M_{n-1m}^{nm}} M_{nm}
\end{array}
$$

$$M_{\bullet m-1}^{\bullet m} \uparrow \quad M_{0m-1}^{0m} \uparrow \qquad M_{1m-1}^{1m} \uparrow \qquad\qquad M_{nm-1}^{nm} \uparrow$$

$$\vdots \qquad\qquad \vdots \qquad\qquad\qquad \vdots$$

$$
M_{\bullet 2}^{\bullet 3} \uparrow \quad M_{12}^{13} \uparrow \qquad M_{22}^{23} \uparrow \qquad\qquad M_{n2}^{n3} \uparrow
$$

$$
\begin{array}{ccccccc}
M_{\bullet 2} & M_{12} & \xrightarrow{M_{12}^{22}} & M_{22} & \xrightarrow{M_{22}^{32}} & \cdots & \xrightarrow{M_{n-12}^{n2}} M_{n2}
\end{array}
$$

$$M_{\bullet 1}^{\bullet 2} \uparrow \quad M_{11}^{12} \uparrow \qquad M_{21}^{22} \uparrow \qquad\qquad M_{n1}^{n2} \uparrow$$

$$
\begin{array}{ccccccc}
M_{\bullet 1} & M_{11} & \xrightarrow{M_{11}^{21}} & M_{21} & \xrightarrow{M_{21}^{31}} & \cdots & \xrightarrow{M_{n-11}^{n1}} M_{n1}
\end{array}
$$

There is a graphcode $\mathcal{G}(M_{\bullet i}^{\bullet i+1})$ for every morphism $M_{\bullet i}^{\bullet i+1}\colon M_{\bullet i} \to M_{\bullet i+1}$ between horizontal slices. Thus, we can define the graphcode of the two-parameter persistence module as the union of the graphcodes for all morphsims $M_{\bullet i}^{\bullet i+1}$.

*Example* D.4.

$$
\begin{array}{ccccccc}
M_3 & \Bbbk & \xrightarrow{(1)} & \Bbbk & \xrightarrow{(0)} & 0 & \longrightarrow 0 \\
\phi_2 \uparrow & (1) \uparrow & & (1\ 0) \uparrow & (0) \uparrow & & \uparrow \\
M_2 & \Bbbk & \xrightarrow{\binom{1}{1}} & \Bbbk^2 & \xrightarrow{(1\ 1)} & \Bbbk & \longrightarrow 0 \\
\phi_1 \uparrow & (0) \uparrow & & \binom{1}{0} \uparrow & (1) \uparrow & & \uparrow \\
M_1 & 0 & \xrightarrow{(0)} & \Bbbk & \xrightarrow{(1)} & \Bbbk & \longrightarrow 0 \\
& 1 & & 2 & & 3 & 4
\end{array}
$$

In Example D.1 we already determined the matrix corresponding to the morphism from the first to second slice. Thus, $\mathcal{M}(\phi_1) = \mathcal{M}(\phi)$ for $\mathcal{M}(\phi)$ as in (5). Similarly we obtain the matrix from the second to third slice

$$
\mathcal{M}(\phi_2) = \begin{array}{c} \\ [1,3) \end{array} \begin{pmatrix} [1,4) & [2,3) \\ 1 & 1 \end{pmatrix} \tag{6}
$$

By combining the matrices $\mathcal{M}(\phi_1)$ and $\mathcal{M}(\phi_2)$ we obtain the following graphcode $\mathcal{G}$

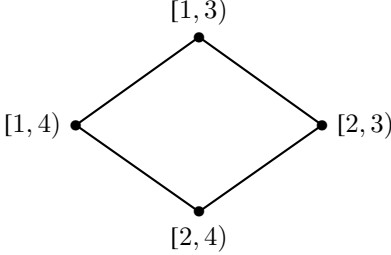

The graphcode is a complete description of a two-parameter persistence module in the following sense: Given the graphcode we can reconstruct the persistence module up to isomorphism. This follows directly from the construction.

Finally we note that we don't necessarily have to restrict to $\mathbb{Z}_2$ coefficients. If $\Bbbk$ is an arbitrary field we can define the graphcode in a similar fashion as a graph with labeled edges, where the label of an edge records the scalar factor $\lambda$ determining the morphism between the two corresponding interval summands. We can still use graphcodes defined in this way for arbitrary fields $\Bbbk$ as inputs for graph neural networks by using an architecture that allows edge weights.

