# OpenReview forum: "Graphcode: Learning from multiparameter persistent homology using graph neural networks"
_NeurIPS.cc/2024/Conference — NeurIPS 2024 poster_

### Official Review · Reviewer_j5Nq · 2024-06-22

**Soundness:** 3
**Presentation:** 4
**Contribution:** 3
**Rating:** 6
**Confidence:** 4

**Summary:**

This paper introduces graphcodes, biparametric persistence summaries that are empirically efficient to compute yet not topologically invariant. It also provides a dedicated C++ library for calculating graphcodes. Graphcodes are structured as layered graphs; each layer contains vertices representing points in persistence diagrams derived from "horizontal slices" of a bifiltration. Edges connecting these points are induced by the inclusion maps between homology groups across the vertical slices.  The authors propose the use of graphcodes as effective representations of data topology for graph neural networks, which facilitates the learning from topological features in datasets through deep learning approaches.  The performance of graphcodes is empirically tested and benchmarked against other (multi)persistent summaries across three distinct tasks: graph classification, shape classification, and binary classification of samples derived from various random point processes. While graphcodes demonstrate superior performance in shape and binary classification tasks, they do not outperform state-of-the-art methods in graph classification.

**Strengths:**

**Significance**:

The paper tackles an important problem in applied topological data analysis: the development of topological summaries or representations that "characterize" (or provide critical information about) multiparameter persistence modules in such a way that they are easy to use/study, as persistence diagrams and their vectorizations do in one-dimensional persistent homology. More than that, the proposed summaries, called graphcodes, are not only good representations in the sense that they are easy to understand (stacked and connected persistence diagrams) as one-dimensional persistence diagrams, but are also easy to represent and manipulate in machine learning tasks, as are diagram vectorizations in the one-dimensional case. The graph nature of the representation makes graphcodes ideal for machine learning tasks, where topological information can be combined with graph learning architectures to automatically extract topological information about the input. This last point may be really relevant in topological deep learning as an automatic topological feature extractor from topological domains.

**Originality**:

Although I am not an expert on the specific topic of multiparameter persistent homology, I know that its integration with machine learning is still in its beginning, and there are not a lot of papers exploring how to create efficient and ML-suitable representations of the topological information of data. This one does, and it does so in a very original and understandable way. Seeing biparametric filtrations as graphs is something clever that I have not seen before and opens the door to automatically extracting topological information from the input data using graph learning methods.

**Quality and Clarity**:

Regarding quality, I find the main text (and the parts of the appendix I read) to be of very good quality. For an expert in topological data analysis, the paper can be followed smoothly, and the main concepts are illustrated with figures. For a non-expert, the theoretical part will be probably hard, but I cannot think of any way of making it clearer with the limited number of pages. Regarding the experiments, the number of tasks is enough for a theoretical paper.

**Weaknesses:**

The main weakness of the paper is that the topological summary is not a topological invariant and depends on a choice of basis. This likely means that, in a real scenario, neural networks fed with graphcodes would require numerous examples to learn how to overcome this limitation. In a context where equivariance and invariance are becoming increasingly important in neural networks to avoid issues like this one, this may prevent graphcodes from being a suitable choice in deep learning pipelines. However, something similar occurs with graph eigenvector positional encodings, yet they are still used in state-of-the-art graph neural networks. The suitability of graphcodes may depend on how sensitive the performance of neural networks trained with graphcodes is to the choice of basis. Given that this conference is oriented towards machine learning, I believe that experiments assessing this sensitivity are crucial for the paper's acceptance.

Regarding the entire set of experiments, I feel the discussion on the graph classification problem is not fair. For instance, it is mentioned that the performance of their methods is affected by small training sets for GNN architectures, but for the proteins dataset, a GNN achieves an accuracy of 84.91%, which is significantly higher than the methods reported, according to PapersWithCode: https://paperswithcode.com/sota/graph-classification-on-proteins. Although it is a drawback that the results with graphcodes are inferior to their counterparts, I do not think this is a critical weakness, and I believe it is preferable to have a more critical discussion of the method. Similarly, I think that the comparison with other methods in graph classification is unfair because they do not report the accuracies of neural networks trained with the other summaries. Although you mention partial experimentation, it is not specified adequately how these experiments were conducted, how many parameters they used, etc., to dismiss the experiments and present results from other tables.

Regarding the text, it is unusual for me to work with half-open intervals [a, a). As stated in Figure 2, some cycles can be born at the same time they die. While it is useful for computations to have these bars, it is not standard to consider persistence diagrams with only some diagonal points. Even the notation is counterintuitive (what does an interval like this signify?). Perhaps it would be worth adding a comment about this fact.

There is a typo in line 134 (bais -> basis).

I believe there is a typo in line 143: I think you meant to say that it "contains consistent bases for all chain vector spaces $Z_p(K_1)$, ..., $Z_p(K_n)$"; subcomplexes are not vector spaces.

Finally, regarding literature review, I feel like the Mapper paper [1], and [2] and [3] could be good additions to the section, The first one is used usually to generate graph representations of datasets that "preserves" topology and can also be used as inputs to graph neural networks. The second one analyzes expressivity of topology for graph learning (coincides very well with the topic). The third one is a topological layer based on persistent homology for graphs.

[1] Singh, Gurjeet, Facundo Mémoli, and Gunnar E. Carlsson. "Topological methods for the analysis of high dimensional data sets and 3d object recognition." PBG@ Eurographics 2 (2007): 091-100.
[2] Rieck, Bastian. "On the expressivity of persistent homology in graph learning." arXiv preprint arXiv:2302.09826 (2023).
[3] Horn, Max, et al. "Topological graph neural networks." arXiv preprint arXiv:2102.07835 (2021).

PS: Some basic experiments (really basic) on sensitivity and a slight modification of the discussion for graph classification being more critique would increase my score to weak accept. More critical experiments on sensitivity, or/and a good discussion about this section may likely make me to increase the score even higher.

**Questions:**

- Vertices of the graph are given by persistence diagrams computed from "horizontal" slices, and edges from vertical morphisms. Does the construction depend on which parameter you select to be "horizontal" and which parameter you select to be "vertical"?

- Can computations be parallelized? If so, what is the impact on the computation time? I saw that you compute the whole graph reducing one big matrix. However, it seems to me that parallelization, like reducing several matrices at a time, could be worth in this context.

- Could this be extended to $n$-parametric persistent homology?

- Are there differentiability results for these computations? You compare your method with PersLay. One of the good points of PersLay is precisely that backpropagation can be computed using PersLay as in intermediate layer due to differentiability results of one-dimensional persistence barcodes. Can something be said about this?

**Limitations:**

Authors have addressed adequately the limitations of their paper through the text.

---

> ### Author Rebuttal · Authors · 2024-08-05
>
> We thank the reviewer for his comments.
>
> **Weaknesses:** Please also see "Expressivity and non-invariance of graphcodes" in the overall rebuttal above.
>
> Given a fixed size dataset one can create multiple graphcodes corresponding to different choices of bases for each instance. This is a way to teach the neural network to learn relevant information independent of the chosen basis without the need of a bigger initial dataset.
>
> We only included the TUDatsets as they seem to be the standard benchmark for multi-parameter topological descriptors. We are aware that GNN's outperform topological descriptors on these datasets and we don't see any evidence that these datasets are particularly well suited for topological descriptors. But we want to point out that graphcodes are not inferior to other methods on all of these datasets. They are certainly not dominant but on some of the datasets they outperform some of the other methods. And non of the methods outperforms every other method on every dataset. One can also not directly compare the data required for a GNN taking the graphs directly as an input to a GNN taking the graphcodes as an input as the graphcodes can be much bigger than the initial graphs.
>
> We actually did experiments testing the sensitivity to different choices of basis. In the uploaded software we implement an option (do-exhaustive-reduction) to compute a basis based on minimal lexicographic cycles and an option (do-random-shuffle) to make random changes to the basis. The minimal lexicographic cycles didn't change the classification accuracy while the random bases lead to slightly worse results. We are happy to include these experiments in a final version. We also propose to change the basis of each graphcode in every iteration of the training process to teach the neural network to extract the relevant information independent of the bases. The problem is that randomly changing the basis is not trivial. The simple method we implemented in (random shuffle) increases the number of edges a lot. Doing this kind of basis changes in every training iteration would make the training process slow. But this is certainly something that can be improved in future work.
>
> **Question 1:** Graphcodes can be constructed by taking slices vertically or horizontally which  we have also implemented in the software. Both choices of direction lead to differtent graphs but both of them contain the same complete information of the two-parameter persistence module. One could also consider taking diagonal slices.
>
> **Question 2:** Yes one could compute the subgraph for any two consecutive slices independent of each other in parallel. For computing graphcodes from arbitrary two-parameter persistence modules that would probably be the most efficient option. But in the case of persistence modules coming from bifiltered simplicial complexes every simplicial slice is contained in the following one. What makes the graphcode algorithm so efficient is that we don't have to compute the persistence of all slices successively but we can compute the graphcode of all slices at once in the same asymptotic time as computing the persistence of the top slice.
>
> **Question 3:** This could theoretically by extended to persistence modules over any poset of the form $P\times \mathbb{R}$ where the $\mathbb{R}$ part is compressed by taking barcodes and maps between bars over different points in $P$ are connected by edges in a similar fashion.
>
> **Question 4:** This is an interesting question but unfortunately we don't have an answer at the moment.

---

> > ### Comment · Reviewer_j5Nq · 2024-08-09
> >
> > Thank you very much for your answers. I'm really glad that you performed experiments on the sensitivity of the method to different choices of basis. I would really love to see a brief discussion of this in the paper. That said, I'm not convinced by your first argument: I need data that corroborate your point. Although theoretically valid, I would need to see if networks are really able to extract information independent of the choice of basis. For example, experiments comparing accuracies with respect to the number of "augmented" representations with different bases would be really clarifying.
> >
> > Regarding datasets, I'm not complaining about the results, but about how the discussion goes in the paper. I'm planning to raise my score, but I would need to see a more critical discussion. As you say, you are starting a new research topic! In particular, I'm talking about sentences like this line:
> >
> > *Moreover, we believe that the performance of our pipeline relative to the other methods is worse on these datasets compared to the following experiments because the training sets are rather small, which is disadvantageous for our GNN architecture*.
> >
> > Also, for improved transparency on the results, I would add some results for usual graph neural networks (non-topological), and discuss the differences among topological and non-topological methods. If you think that it is really the case that this will work with more examples better than other SOTA methods, then I would require you to do experiments with bigger datasets, like ogb-lsc: https://ogb.stanford.edu/docs/lsc/pcqm4mv2/
> >
> > This said, I'm really excited about this avenue of multiparameter persistent homology and machine learning. I'm going to trust you will make the changes requested and I'll raise my score to 6. Also, please address the comments I made that were not reflected in your answer (e.g., the ones about the typos).
> >
> > P.S.: Could you please include a reference for the theorem in the general rebuttal? I'm interested in that theorem and its formal formulation.

---

### Official Review · Reviewer_1X25 · 2024-07-11

**Soundness:** 2
**Presentation:** 2
**Contribution:** 2
**Rating:** 4
**Confidence:** 4

**Summary:**

This paper introduces "Graphcode", a new representation for summarizing the topological properties of datasets filtered along two parameters. Graphcodes are based on persistent homology but aim to provide a more interpretable and efficient summary than existing multi-parameter topological descriptors. The key idea is to collect one-parameter sliced barcodes with basis-choice dependent mappings between consecutive persistence diagrams, which results in a bipartite graph structure, as the so-called Graphcode summary.

The authors present an efficient algorithm to compute Graphcodes and demonstrate how they can be directly used as input to graph neural networks for machine learning tasks. Experiments on several datasets show that Graphcodes have competitive or superior performance compared to other existing topological descriptors of multi-parameter persistence, especially on tasks with clear topological signals.

**Strengths:**

- A new topological representation that captures two-parameter topological information.
- Efficient computation algorithm that scales well to large datasets.

**Weaknesses:**

- Graphcode are not topologically invariant, depending on choice of basis. Impact of this is not fully explored.
- Theoretical analysis of expressiveness/information captured by Graphcode is limited.

**Questions:**

- How sensitive are the results to the choice of basis used in constructing the Graphcode? Is there a way to make this choice optimal or at least consistent/stable locally?
- Since the Graphcode is not topological invariant, one concern might be that if the GNN based on such method is still permutational invariant / equivalent? If yes, more discussion or justification should be included in the main text. If not, the potential issue or constraints in application should be discussed.
- The theoretical foundations and analysis of what information Graphcode capture compared to other descriptors is somewhat limited. Can this be expanded?

**Limitations:**

As the author already mentioned in the paper, one main concern is raised from the issue that the Graphcode, as a topological descriptor, is not invariant. It might also break the stability property of persistence modules. A GNN based on Graphcode might no longer be permutation invariant or equivalent. More discussion about such limitations in both theory and application will buy the paper more benefits.

---

> ### Author Rebuttal · Authors · 2024-08-05
>
> We thank the reviewer for his comments.
>
> **Question 1:** The standard graphcode algorithm chooses a basis based on the reduction performed by the slicewise persistence algorithm. So in some sense the algorithm chooses a specific basis. In the uploaded software we also added an option  (do-exhaustive-reduction) to compute a basis based on minimal lexicographic cycles and an option (do-random-shuffle) to make random changes to the basis. The minimal lexicographic cycles didn't change the classification accuracy while the random bases lead to slightly worse results. The problem with the way we perform random basis changes is that it strongly increases the number of edges. This can be explained by the way the reduction in the persistence algorithm works which leads to a relatively small graph. But we think that there is room for improvement in shuffling the bases without massively increasing the number of edges. But overall changing the bases did not dramatically affect the results.
>
> **Question 2:** The permutation invariance of graph neural networks is a feature of the architecture itself and is true independent of the input graph. But if the question is if two different graphs representing the same persistence module can lead to different classification results then theoretically this can happen. One way to deal with this problem is to randomly change the basis of the graphcode in every iteration of the training process to teach the neural network to extract the ismorphism type of the persistence module independent of the basis. As explained in the previous paragraph the current implementation of this random basis change increases the number of edges too much which makes the whole process slow. But this is something that can be worked out and we are optimistic that especially the expertise of the machine learning community can help a lot to improve the architecture.
>
> **Question 3:** Please see "Expressivity and non-invariance of graphcodes" in the overall rebuttal above.

---

> > ### Comment · Reviewer_1X25 · 2024-08-14
> >
> > Thank you answering the questions. I will keep my score.

---

### Official Review · Reviewer_NxrC · 2024-07-12

**Soundness:** 2
**Presentation:** 2
**Contribution:** 1
**Rating:** 5
**Confidence:** 3

**Summary:**

The paper introduces "graphcodes," a novel multi-scale summary of the topological properties of datasets using graph neural networks. Unlike traditional persistent homology, which uses a single parameter, graphcodes handle datasets filtered along two real-valued scale parameters, resulting in a more informative and interpretable summary. The paper outlines how graphcodes can be efficiently computed and integrated into machine learning pipelines, demonstrating improved classification accuracy over state-of-the-art approaches.

**Strengths:**

The paper introduces graphcodes, extending persistent homology to two-parameter scales, offering a novel and efficient method that integrates seamlessly into machine learning pipelines via graph neural networks. It demonstrates superior classification accuracy on various datasets compared to existing methods and provides an interpretable summary of topological features. The approach claims efficient computation comparable to one-parameter summaries, adding value through its innovation and practical performance.

**Weaknesses:**

From the experimental results, it’s hard to find too much practical meanings of using graphcode.

**Questions:**

1.	How is the graph constructed in graphcode? Is it a directed graph?
2.	From Table 1, MP-HSM-C performs much better than GC. Moreover, as stated in Line 251, all approaches terminate within a few seconds even on the devices mentioned in Line 240. So do we really need a method that runs a few seconds faster at the cost of losing considerable accuracies?
3.	As stated in Line 227, GC-NE can be viewed as Perslay. It seems that Perslay is already accurate and efficient enough. From my perspective, it’s more practical to use Perslay rather than graphcode considering the experimental results provided in this paper. Besides, how does GC-NE perform on graph classification tasks?
4.	As mentioned in line 172, the graphcode depends on the chosen barcode bases. I wonder if there is a risk that different choices of barcode bases could impact the robustness and consistency of GC?

**Limitations:**

See Questions

---

> ### Author Rebuttal · Authors · 2024-08-05
>
> We thank the reviewer for his comments.
>
> **Question 1:** The construction of the graph is explained in Section 3, Appendix B and D. The vertices are the points of the persistence diagrams (topological cycles) of the individual slices where two points in consecutive slices are connected by an edge if the representative cycle of one point maps to the representative cycle of the other point. See Figure 3 for a schematic depiction. We only use undirected edges but using directed edges pointing in the direction of the filtration is certainly something that can be explored in the future.
>
> **Question 2:** Multi-parameter persistent homology especially in combination with machine learning is still in its infancy. Only recent advances in algorithms made it possible to apply these methods on larger datasets. Therefore, there is still a lack of good benchmark datasets for multi-parameter topological descriptors. We used the TUDataset for comparison because it was the most common dataset in the related literature on combining multi-parameter persistent homology and machine learning but we don't see any evidence that topological descriptors are particularly well suited for these datasets. We believe that these datasets where chosen mainly because of their small size. It is true that all approaches terminate within a few seconds but for example the MUTAG dataset has only 188 samples. As shown in Table 2, on larger datasets other methods took significantly longer than graphcode.
>
> **Question 3:** The original Perslay architecture is constructed for one-parameter persistent homology. So GC-NE is not exactly the same as Perslay. To our knowledge the adaption of Perlay to two-parameter persistent homology has also not been considered before. A stronger topological descriptor is only beneficial in the presence of enough topological signal. On some datasets weaker topological descriptors will be sufficient and the edges will not contribute much to the accuracy. But for example on the shape dataset in Table 2 the edges improve the accuracy by 4.1\% at basically the same computational cost.
>
> **Question 4:** Please see "Expressivity and non-invariance of graphcodes" in the overall rebuttal above.

---

> > ### Comment · Reviewer_NxrC · 2024-08-12
> >
> > If there is no evidence that topological descriptors are particularly well suited for these datasets, what is making Graphcode better than compared methods?Besides, I cannot find a clear response to Question 4.

---

> > > ### Author Response · Authors · 2024-08-12
> > >
> > > We do not claim that graphcodes are better compared to other topological descriptors on the TUDatasets. They outperform most of the other methods on some of the TUDatasets and they don't do well on others. If you look at this table you see that there is no method that outperforms all the other methods. But on all the datasets we considered, except the TUDatasets, graphcodes outperformed all the other methods in computation time and classification accuracy. Although these might not be "real world" datasets, given their more topological/geometric nature, it is still a strong proof of concept that graphcodes are very good at picking up the topology or geometry of datasets. Of course one can ask the broader question of how useful are topological methods on real world datasets in general? But as mentioned before topological data analysis is a young and developing field and it can take time for methods to be successfully applied in practice. It also took quite some time for neural networks to be successfully applied in practice.
> > >
> > > **Regarding question 4:** Theoretically there is a risk that the basis dependence can effect the robustness and consistency. But in practice they still perform well on the geometric datasets. We propose to deal with this dependence on representation by showing the neural network graphcodes with respect to different choices of basis for each instance to teach it to extract the relevant information independent of the basis. That such an approach can work is demonstrated, for example, in the lastest alphafold paper https://www.nature.com/articles/s41586-024-07487-w  where they got rid of the equivariance and invariance constraints of the previous version. We implemented options in our graphcode software package to compute an alternative basis based on minimal lexicographic cycles and an option to randomly change the basis. The minimal lexicographic cycle bases did not effect the results at all while the random bases did slightly worse. There are still problems with the random change of basis as it strongly increases the number of edges but there are many avenues for future research to improve the pipeline.

---

> > > > ### Comment · Reviewer_NxrC · 2024-08-13
> > > >
> > > > > The minimal lexicographic cycle bases did not effect the results at all while the random bases did slightly worse. There are still problems with the random change of basis as it strongly increases the number of edges but there are many avenues for future research to improve the pipeline.
> > > >
> > > > Is there any supporting evidence? Or is it considered as a limitation in this paper?

---

> > > > > ### Author Response · Authors · 2024-08-13
> > > > >
> > > > > Regarding the first sentence: We are happy to include experiments with minimal lexicographic cycle and random bases in a final version.
> > > > >
> > > > > Regarding the second sentence: To establish a learning pipeline where in each iteration of the training process the bases of the graphcodes are randomly shuffled we need an efficient method of doing random basis changes without blowing up the graphs. This is still work in progress and is not part of this paper but it is an obvious way to deal with the basis dependence in future work.

---

> > > > > > ### Comment · Reviewer_NxrC · 2024-08-13
> > > > > >
> > > > > > Thank you for your response. I understand that it's hard to include additional experiments now, but I still believe that comparisons with random bases is necessary in revealing the stability of the proposed method. I have revised my score.

---

### Official Review · Reviewer_k1Z5 · 2024-07-12

**Soundness:** 3
**Presentation:** 3
**Contribution:** 3
**Rating:** 7
**Confidence:** 5

**Summary:**

This paper proposes a computationally fast method to extract information from a 2-parameter persistence module. The authors consider a 2-parameter persistence module as slices of 1-parameter persistence modules. Each 1-parameter persistence module can be represented as barcodes. The authors consider these barcodes as basis and represent the maps between the barcodes, induced by the bifiltration, as an attributed graph. This attributed graph is used as an input to a GNN architecture. The information extracted by this method is not a topological invariant, however, empirical results show that the method gives comparable results on TU datasets and better results on some synthetic datasets as compared to existing multiparameter methods.

**Strengths:**

1. Overall, the paper is well-written and organized.

2. The problem that the authors are trying to tackle is a hard one of capturing relevant information in 2-parameter persistence in the absence of a complete invariant.

3. The method proposed in the paper has substantial theoretical backing and an algorithmic component.

4. The proposed method is computationally fast, as indicated by results in Table 2.

**Weaknesses:**

1. The authors have shown experiments on TU Datasets and some synthetic datasets. GC seems to perform well on the synthetic datasets, however, the performance on TU datasets is not particularly impressive. The authors claim that it might be because there are not enough topological signals to capture in those datasets, which is not fully convincing.

2. The exact experimental setup (hyperparameter choices, number of GAT layers etc.) is not described in the paper, not even in the appendix.

**Questions:**

1. Was there a specific reason to choose GAT as the GNN architecture? How does GC perform with other GNN architectures?

2. Was there a specific reason to choose max-pooling? How does the model perform with sum/average pooling?

3. It is interesting to see from Table 3 and Table 4 that adding edges in GC amounts to a marginal increase in performance. Does this mean that the information carried by the maps between barcodes is not as important?

4. How does the proposed method stand with respect to the expressivity as compared to other multiparameter persistent homology methods?

**Limitations:**

Yes, the authors have discussed limitations.

---

> ### Author Rebuttal · Authors · 2024-08-05
>
> We thank the reviewer for his comments.
>
> **Question 1 and 2:** The idea of choosing a GAT network is that topological (homological) features that are persistent across both scale parameters are reasonable candidates for the topological signal and that the network should learn to pay more attention to the features of high persistence in consecutive slices that are connected by an edge. We also tested standard graph convolution networks and average pooling which both performed worse. We want to point out that the proposed architecture is meant to serve as a proof of concept and we don't want to claim that it is close to optimal. There is certainly a lot of room for improvement and we hope that especially the expertise of the machine learning community will help to advance this architecture.
>
> **Question 3:** The graphcodes are very strong topological descriptors but this expressivity comes at the cost of basis dependence. We believe that this tradeoff is more favourable for graphcodes if there is more topological signal present in the data. The random point process data does not really have a significant topological signal and also many of the Orbit point clouds look very much like random noise. Therefore, the edges of the graphcode don't contribute as much as they do in the shape dataset. But for the Orbit100k dataset they add another 0.8\% which might be significant given the already very high classification accuracy. Also on the TUDatasets, given the small size of these datasets and the at least questionable topological signal, we think that the tradeoff between expressivity and (basis) consistency becomes unfavourable for graphcodes. But we want to point out that, although graphcodes are not dominant on the TUDatsets, on some of those datasets graphcodes still outperform some of the other methods.
>
> **Question 4:** Please see "Expressivity and non-invariance of graphcodes" in the overall rebuttal above.

---

> > ### Comment · Reviewer_k1Z5 · 2024-08-08
> >
> > I would like to thank the authors for their response. I have read the responses and would like to stick to my score.

---

### Author Rebuttal · Authors · 2024-08-05

At first, we want to thank all reviewers for their comments. As some points came up in more than one review we want to address them here in a general rebuttal and address more specific questions in individual rebuttals below.

**Expressivity and non-invariance of graphcodes:** The most important point we want to address is the critique of non-invariance of graphcodes which came up in all of the reviews. It is a well-known theorem in the field of topological data analysis that there can not exist a complete (capturing all the information) and discrete invariant of multi-parameter persistence modules. Some reviewers asked about the expressivity or the information captured by graphcodes. The graphcode of a two-parameter persistence module is a complete descriptor which means that it captures all the information of the two-parameter persistent homology up to isomorphism. It is easy to reconstruct the initial module up to isomorphism from the graphcode. It is just an alternative compact representation of the module that is suitable as input for graph neural networks. Other established methods construct a discrete invariant from a module which comes at the cost of significant loss of information. The theorem mentioned before implies that one is forced to make this tradeoff between invariance and loss of information and the lack of invariance of graphcodes has to be put into this context. From this perspective, on appropriate datasets (like the shape dataset), the lack of invariance is rather a strength than a shortcoming since it is the only way we can capture all the information. A two-parameter persistence module can be equivalently represented by two graphs with different edges (the vertices and vertex labels are actually invariant) but both graphs contain exactly the same information.

**Graphcodes as inputs for neural networks:** Graphcodes are a novel approach to combine multi-parameter persistent homology and machine learning by giving up on invariance for the benefit of providing complete two-parameter persistence information to a neural network. Since this has not been considered before and as the reviewers have pointed out there are a lot of open questions like: How well can a graph neural network architecture learn to extract the isomorphism type of the underlying module from the graphcode and use this information to make predictions? What is the best architecture to learn from graphcodes? How to efficiently change the bases of the graphcodes in the learning process to teach the neural network to extract the isomorphism type of a module independent of the basis?  As acknowledged by all the reviewers, we built the theoretical foundation of graphcodes, an efficient algorithm to compute graphcodes and proposed an architecture that works well on some datasets which provides a proof of concept. Hence, we think that these open questions are not a reason to reject the paper. Every line of research has to start at some point. Since most of these open questions are on the machine learning side of the pipeline we believe that it would be very beneficial to bring this work to the attention of the machine learning community.

---

### Decision · Program_Chairs · 2024-09-25

**Decision:**

Accept (poster)

**Comment:**

This paper presents a new multi-scale summary of topological features of graphs, called a *graphcode*. Graphcodes are obtained from bifiltrations, i.e. a filtration process along two (independent) variables on a graph. Leveraging ideas from computational topology, in particular persistent homology, the paper calculates graphcodes as a representation of maps between consecutive slices in a stack of persistence diagrams, i.e. topological descriptors. The resulting features can be easily integrated into a deep-learning model, leading to a new way of solving graph-learning tasks.

Initial support of the paper was not very pronounced; reviewers agreed on the utility, novelty, and relevance of the idea, but cited concerns about the experimental setup as well as some of the theoretical properties. For the latter point, the fact that graphcodes—unlike many other methods in topology-based graph learning—are *not* a topological invariant was raised multiple times. The authors acknowledge this fact openly in their submission, though, and the AC believes that this is not a shortcoming *per se*, since graphcodes can nevertheless be gainfully employed empirically (and most, if not all, graph-learning methods do not even discuss properties like invariance). That being said, the other shortcoming, raised by multiple reviewers, concerns the experimental results and their framing. For a new method like this, which the AC construes to be primarily about showing that multi-parameter persistent-homology methods can be computationally efficient, a paper cannot be expected to report state-of-the-art results. Nevertheless, the AC essentially echoes the feedback by reviewer `j5Nq`, who made additional suggestions about (a) discussiong **additional related work**, and (b) an **improved contextualisation of the results**.

Since the authors engaged openly and actively in their rebuttal, several of the concerns could already be alleviated. Thus, the authors are trusted to implement the aforementioned suggestions, as well as other suggestions raised by reviewers (mostly concerning the accessibility of the work), in the camera-ready version of their paper.